# How prolonged expression of Hunchback, a temporal transcription factor, re-wires locomotor circuits

**Julia L Meng[1,2], Zarion D Marshall[1], Meike Lobb-Rabe[1,2], Ellie S Heckscher[1]***

[1]Department of Molecular Genetics and Cell Biology, Grossman Institute for Neuroscience, Program in Cell and Molecular Biology, University of Chicago, Chicago, United States; [2]Program in Cell and Molecular Biology, University of Chicago, Chicago, United States

**Abstract** How circuits assemble starting from stem cells is a fundamental question in developmental neurobiology. We test the hypothesis that, in neuronal stem cells, temporal transcription factors predictably control neuronal terminal features and circuit assembly. Using the Drosophila motor system, we manipulate expression of the classic temporal transcription factor Hunchback (Hb) specifically in the NB7-1 stem cell, which produces U motor neurons (MNs), and then we monitor dendrite morphology and neuromuscular synaptic partnerships. We find that prolonged expression of Hb leads to transient specification of U MN identity, and that embryonic molecular markers do not accurately predict U MN terminal features. Nonetheless, our data show Hb acts as a potent regulator of neuromuscular wiring decisions. These data introduce important refinements to current models, show that molecular information acts early in neurogenesis as a switch to control motor circuit wiring, and provide novel insight into the relationship between stem cell and circuit.

DOI: https://doi.org/10.7554/eLife.46089.001

**\*For correspondence:**
heckscher@uchicago.edu

**Competing interests:** The authors declare that no competing interests exist.

## Introduction

In complex brains, a relatively small pool of neuronal stem cells generates a large diversity of neurons. Ultimately, this diversity of neurons wire together to generate functional neuronal circuits that underlie sensation, cognition, and action. How circuits are assembled starting from neuronal stem cells is a fundamental question in developmental neurobiology that has implications for the fields of biomedicine and evolution.

As a neuronal stem cell divides it generates a diverse set of neurons in a predictable, orderly fashion through stem cell-intrinsic molecular programs. This process is often referred to as temporal patterning. In past decades, intense molecular genetic work has shown that early step of temporal patterning—the establishment of unique profiles of gene expression in newly-born post mitotic neurons—can be controlled by a class of transcription factors called temporal transcription factors (*Doe, 2017*). Despite decades of work on temporal transcription factors, however, the biological relevance of most published manipulations is unknown. This is because the role of temporal transcription factors at later steps of neuronal development—establishing terminal neuronal features such as dendrite morphology, axonal trajectory, and functional synaptic partnerships—are almost completely unexplored.

Temporal patterning has been proposed to play a role in circuit assembly (*Deguchi et al., 2011*). An attractive and often cited model is that temporal transcription factors are master regulators of entire developmental programs (*Allan and Thor, 2015*). From a circuit wiring perspective, this would mean that manipulating temporal transcription factor expression in a given stem cell would be

sufficient to predictably alter the wiring of neuronal progeny. If this is true, it has important implications for the evolution of neuronal circuits and for stem-cell based biomedical interventions. In support of such a model, there is an association between neuronal birth time and neuronal circuit membership in many organisms and many brain regions (*Bhansali et al., 2014*; *Deguchi et al., 2011*; *Eerdunfu et al., 2017*; *Greaney et al., 2017*; *Jefferis et al., 2001*; *Kulkarni et al., 2016*; *McLean et al., 2007*; *McLean and Fetcho, 2009*; *Morrow et al., 2008*; *Osterhout et al., 2014*; *Petrovic and Hummel, 2008*; *Pujol-Martí et al., 2012*; *Tripodi et al., 2011*). However, such a simple model over emphasizes the idea that temporal transcription factors are master regulators of temporal cell fate and de-emphasizes the role of dynamic factors in the environment or in the stem cell that change as temporal patterning occurs. These other dynamic factors could also play a major role in circuit wiring. Thus, it is far from clear how manipulating temporal transcription factors impacts circuit assembly.

The objective of this study is to test the hypothesis that temporal transcription factor activity in neuronal stem cells is sufficient to control neuronal terminal features and therefore circuit assembly. The Drosophila embryonic/larval nerve cord is a model system ideally suited to study how temporal transcription factors regulate circuit wiring. Here, we focus on one stem cell, neuroblast NB7-1, and five of its progeny, the U motor neurons, U1-U5 (*Figure 1A*). Molecular markers are available to uniquely identify each U motor neuron (*Figure 1A*), and the temporal transcription factors involved in establishing U motor neuron diversity have been studied for decades (*Grosskortenhaus et al., 2006*; *Isshiki et al., 2001*; *Pearson and Doe, 2003*). For example, one of the first-identified and best-characterized temporal transcription factors, Hb (Hb), is involved in establishing early-born U1/U2 motor neuron identities (*Isshiki et al., 2001*). Notably, the mammalian homolog of Hb, Ikaros, acts as a temporal transcription factor in both retina and cortex (*Alsiö et al., 2013*; *Elliott et al., 2008*). In this study, to use the Drosophila motor system, and specifically NB7-1 and U motor neurons, to study the role of temporal transcription factors in circuit wiring, first we characterize each U motor neuron's terminal features—dendrite morphology, axonal trajectory, and neuromuscular synaptic partner—and link these features to U motor neuron marker gene expression at single neuron resolution. Next, we manipulate temporal patterning in NB7-1 using three different Hb manipulations: we prolong the expression of wild type Hb, we prolong expression of an activating Hb protein (tethered to the VP16 transcriptional activation domain), and we shorten the expression of wild type Hb by precocious expression of the switching factor, Seven-up (*Kanai et al., 2005*; *Tran et al., 2010*). In these backgrounds, we monitor establishment of U motor neuron molecular identity (i.e., embryonic marker gene expression), as well as maintenance (i.e., larval marker gene expression), and assess U motor neuron terminal features.

Our data uncover the biological relevance of manipulating temporal transcription factors. We find that Hb is a potent regulator of neuromuscular wiring decisions. But, surprisingly, we find prolonged expression of Hb leads to a transient rather than permanent specification of U motor neuron molecular identity, and that the embryonic molecular markers which have been used for decades to monitor U motor neuron embryonic molecular identity cannot accurately predict U motor neuronal terminal features. Nonetheless, our data support a general model in which neuromuscular synaptic partner choice and circuit wiring can be established before neuronal birth by early programs of gene expression acting in neuronal stem cells.

## Results

### Characterizing terminal features of U motor neurons at single neuron resolution

We take advantage of a well-characterized neuronal stem cell (neuroblast) lineage in the Drosophila embryonic larval nerve cord to test the hypothesis that temporal transcription factor activity controls neuronal terminal features and circuit assembly. Specifically, we focus on NB7-1, which generates five U motor neurons in this order: U1, U2, U3, U4, U5, followed by a series of poorly-characterized neurons (*Figure 1A*). U motor neurons are easily identified because they express the homeobox transcription factor, Even-skipped (Eve, Evx1/2 in vertebrates), and in embryos each U motor neuron can be uniquely identified by a distinctive combination of transcription factors (*Figure 1A*) (*Isshiki et al., 2001*). Furthermore, each U motor neuron occupies a distinctive position within the

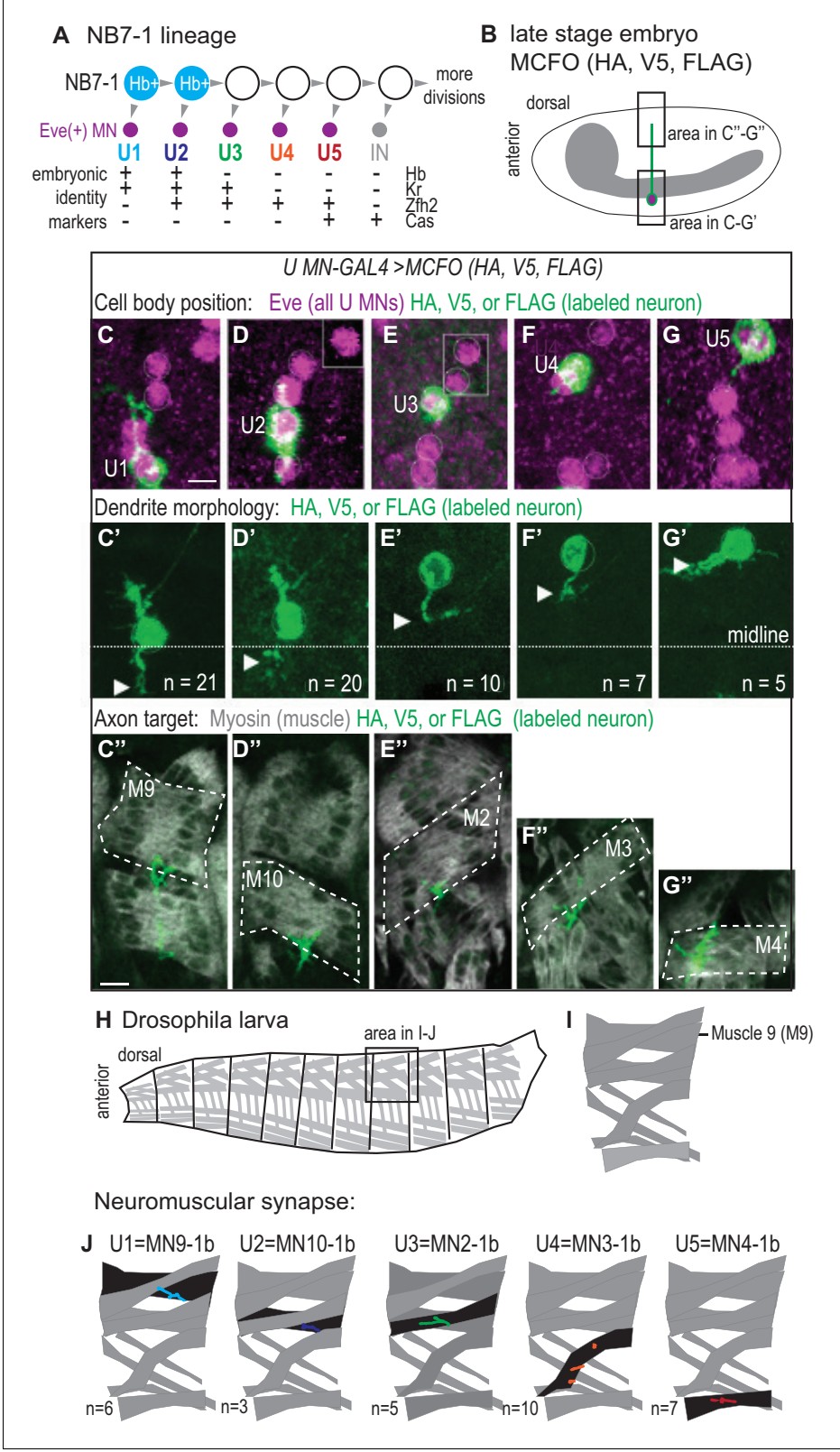

**Figure 1.** U motor neurons have unique embryonic molecular identities and unique terminal features. (**A**) Illustration of the divisions of NB7-1. Each gray arrowhead represents cell division. Abbreviations: MN is motor neuron, IN is interneuron, Eve is Even-skipped, Hb is Hunchback, Kr is Kruppel, Zfh2 is Zinc finger homeodomain 2, and Cas is Castor. (**B**) Illustration of a Drosophila late stage embryo, CNS in gray, motor neuron in green, with

*Figure 1 continued on next page*

*Figure 1 continued*

cell body in magenta. (**C–G''**) Images of individually labeled U motor neuron cell bodies in the CNS of late stage embryos. The same cell is shown in ventral view (**C–G'**) or lateral view (**C''–G''**). Multi-Color Flip Out (MCFO) transgenes were used to stochastically label neurons within a GAL4 pattern with membrane tethered epitope tags (HA, V5, FLAG). (**C–G**) Boxes are insets from a different focal plane. (**C'–G'**) U1/U2 motor neurons dendrite tips (arrowhead) cross the midline (dotted white line), whereas U3-U5 neurons do not. n = number of single-labeled Eve(+) cells. (**C''–G''**) Axons project to unique dorsal muscles (dotted outline, M stands for muscle, for example M9 is muscle 9). (**H–I**) Illustrations of Drosophila larval muscles. (**H**) The larval body is organized into repeated left-right, mirror image hemisegments. (**I**) In a hemisegment, individual muscles are identified and named by characteristic morphology. (**J**) Illustration of individual U motor neuron neuromuscular synapses onto dorsal muscles in larvae. Embryonic motor neuron (e.g., U1) and larval motor neuron synapse (e.g. MN9-1b) names are shown. n = number of single-labeled Eve(+) cells. Color code as in A. For sample of larval image data see *Figure 1—figure supplement 1*. Images in (**C–G'**) are shown anterior to the left. Scale bars represent five microns (**C'–G'**) and 10 microns (**C''–G''**). MCFO transgenes were driven with a U MN-GAL4 line, *CQ2-GAL4 (hsFLP; CQ2-GAL4/+; UAS(FRT.stop)myr::smGdP-HA, UAS(FRT.stop)myr::smGdP-V5-THSUAS(FRT.stop)myr::smGdP-FLAG/+)*.
DOI: https://doi.org/10.7554/eLife.46089.002
The following figure supplement is available for figure 1:

**Figure supplement 1.** U Motor Neuron axon projection in third instar (**L3**) Drosophila larva.
DOI: https://doi.org/10.7554/eLife.46089.003

---

CNS characteristic of its birth time, with earliest-born U motor neurons found closest to the midline (*Figure 1C*). Cell body position in the Drosophila CNS is roughly correlated with birth time because there is little neuronal migration, and so as laterally positioned neuroblasts divide, later-born neurons displace earlier-born neurons to a deeper and more medial CNS position. The establishment of distinct U motor neuron identities has been intensely studied, but the biological relevance of altering embryonic identity markers is poorly understood (*Doe, 2017*). Here, we characterize terminal features of each U motor neuron in wild type.

First, we label individual U motor neurons in whole mount, late-stage embryos when neuroblasts have stopped dividing and circuit wiring is being established (*Figure 1B*). We find dendrites of the earliest-born U motor neurons, U1 and U2, are contralateral (i.e., cross the midline) and dendrites of U3-U5 are ipsilateral (i.e., do not cross the midline) (*Figure 1C'–G'*). Notably, U1 and U2 are generated from NB7-1 while Hb is expressed in NB7-1 (*Figure 1A*). Therefore these data establish a correlation between Hb expression and U motor neuron dendrite morphology and raise the possibility of a causal link.

We visualize single U motor neuron axons and their muscle targets and find every U motor neuron axon projects to a unique dorsal muscle (*Figure 1C''–G''*). The earliest-born U motor neuron, U1 has the longest axon and reaches the distal most muscle, muscle 9 (M9). Progressively later-born U motor neurons have progressively shorter axons, reaching progressively less distal muscles, with U2 reaching muscle 10, U3 reaching muscle 2, U4 reaching muscle 3, and U5 reaching muscle 4. These data agree with previous reports showing that, in general, U motor neurons project to dorsal (and possibly ventral) muscles (*Bossing et al., 1996*; *Landgraf et al., 1997*; *Schmid et al., 1999*).

Next, we label individual U motor neurons in late stage larvae (L3), at a time when there are large, well-established, well-characterized neuromuscular synapses (*Hoang and Chiba, 2001*). Consistent with embryo data, U1 generates a synapse on muscle 9 that has been called MN9-1b (i.e., <u>m</u>otor <u>n</u>euron that targets muscle 9 and has a type <u>1 b</u>ig synapse), U2 generates the MN10-1b synapse, U3 generates the MN2-1b synapse, U4 generates the MN3-1b synapse, and U5 generates the MN4-1b synapse (*Figure 1H–J* and *Figure 1—figure supplement 1*). We conclude that in wild type there is a perfect correspondence between axonal target and neuromuscular synapse and that a majority of dorsal muscles are innervated by neurons from NB7-1.

Together these data link markers of U motor neuron embryonic temporal identity to U motor neuron terminal features—dendrite morphology, axonal target, and neuromuscular synapses- at single neuron resolution, which allows us to predict which terminal features should be associated with a particular U motor neuron marker gene expression profile, regardless of neuronal birth time.

# Prolonged expression of Hunchback in NB7-1 generates ectopic Eve-expressing neurons with U1 embryonic molecular identities at abnormally late times in development

To test the hypothesis that temporal transcription factor activity in neuronal stem cells is sufficient to control neuronal terminal features and circuit assembly, we manipulate temporal transcription factor expression. In wild type, a given temporal transcription factor is expressed in nearly all neuronal stem cells during a limited time window. For example, in NB7-1, the temporal transcription factor Hunchback (Hb) is expressed during the first two neuroblast divisions (*Figure 2A*). In this study, we prolong Hb expression specifically in NB7-1, and then we assay NB7-1 progeny for embryonic marker gene expression (*Figure 2*), the extent to which progeny are motor neurons (*Figure 3*), dendrite morphology (*Figure 4*), functional neuromuscular synapse formation (*Figures 5,6*), axonal trajectory (*Figure 7*), and larval marker gene expression (*Figure 8*).

Hb expression has been prolonged in NB7-1 in many ways (e.g., *hs-GAL4, En-GAL4, Pros-GAL4*, removal of Hb switching factor SVP; *Cleary and Doe, 2006*; *Grosskortenhaus et al., 2005*; *Isshiki et al., 2001*; *Kanai et al., 2005*; *Pearson and Doe, 2003*), and in general, these manipulations produce a large number of ectopic Eve(+) cells that express U1 molecular markers. Here, we use a previously uncharacterized, NB7-1-specific method to prolong Hb expression. NB7-1-specific manipulation is essential when examining circuit wiring because it rules out possible effects of Hb acting in another lineages. Specifically, we use *NB7-1-GAL4* (*Figure 2—figure supplement 1A–B*) to drive Hb expression from either one or two copies of *UAS-Hb* (*Kohwi et al., 2013*). In both manipulations, we find an average of ten Eve(+) cells (*Figure 2A–D',H*). Notably, however, there is hemisegment to hemisegment variability in how long *NB7-1-GAL4* drives gene expression, which results in variability in the number of Eve(+) cells (*Figure 2H,Q*). We also note driving two copies of Hb generates slightly stronger phenotypes (*Figure 2—figure supplement 1E–G*), and so unless otherwise noted, we drive two copies of Hb, which we refer to as 'NB7−1>Hb'. In comparison to NB7−1>Hb, a similar number of Eve(+) cells are found when Hb expression in NB7-1 is prolonged by eliminating Seven-up, a factor that promotes Hb switching (*Kanai et al., 2005*). This suggests that the level of Hb expression we achieve in the NB7−1>Hb manipulation is in a physiological range.

Because we use a previously uncharacterized manipulation of Hb, we perform a series of control experiments to show that we elicit changes in temporal patterning similar to those described previously. We track all NB7-1 progeny in control and NB7−1>Hb embryos using a GFP reporter (*Figure 2C–I*). First, we find a slight, but not significant increase in the total number of GFP(+) NB7-1 progeny in NB7−1>Hb in comparison to Control, which is not enough to account for the number of extra U motor neurons (*Figure 2G*). Second, in Control, later-born NB7-1 interneurons express the transcription factor, Castor (Cas), but not Eve, and are located dorsolaterally to U motor neurons (*Figure 2E–E',I*). In NB7−1>Hb embryos, there is a near complete loss of the later-born NB7-1 Cas (+) neurons (*Figure 2E–F',I*). Taken together, we conclude that in NB7−1>Hb, extra Eve(+) neurons are generated at the time when neurons with later-born molecular identities are normally born.

Next, we characterize embryonic molecular identities of the Eve(+) cells in NB7−1>Hb using the combination of molecular markers Hb (Hb), Krupple (Kr), and Zfh2, which together label U1 (Hb[+] Kr[+] Zfh2[-]), U2 (Hb[+] Kr[+] Zfh2[+]), U3 (Hb[-] Kr[+] Zfh2[+]), and U4/U5 (Hb[-] Kr[-] Zfh2[+]) (*Figure 2J–Q*). Note, we use Hb in two capacities in this experiment. We manipulate Hb in the neuroblast, and we use Hb as marker for U1/U2 molecular identity because in wild type U1 and U2 motor neurons actively transcribe Hb (*Kohwi et al., 2013*). To distinguish actively-transcribed Hb (an identity marker) from Hb inherited from neuroblast cytoplasm, we stain at late embryonic stages to allow for Hb protein turnover as in Pearson and Kohwi (*Kohwi et al., 2013*; *Pearson and Doe, 2003*). In NB7−1>Hb embryos, 86% of Eve(+) cells are U1-like (*Figure 2P-Q* and *Figure 2—figure supplement 1F-G*). Furthermore, in nearly one-third of all hemisegments all Eve(+) cells have U1 molecular identities (*Figure 2Q* arrows). To demonstrate that in NB7−1>Hb Eve(+) cells with U1 identities are born at abnormally late times in development we measure the distance of each cell body from the CNS midline as a proxy for neuronal birth time, and then plot the position of Eve(+) cells along the medio-lateral axis (*Figure 2P* and *Figure 2—figure supplement 1E*). Together these data demonstrate that prolonging Hb expression with NB7-1-GAL4 is sufficient to generate extra Eve(+) neurons, a majority of which have U1 molecular identities regardless of birth time. Thus, in NB7−1>Hb, a neuron's birth time and its temporal identity are decoupled, which allows us to parse

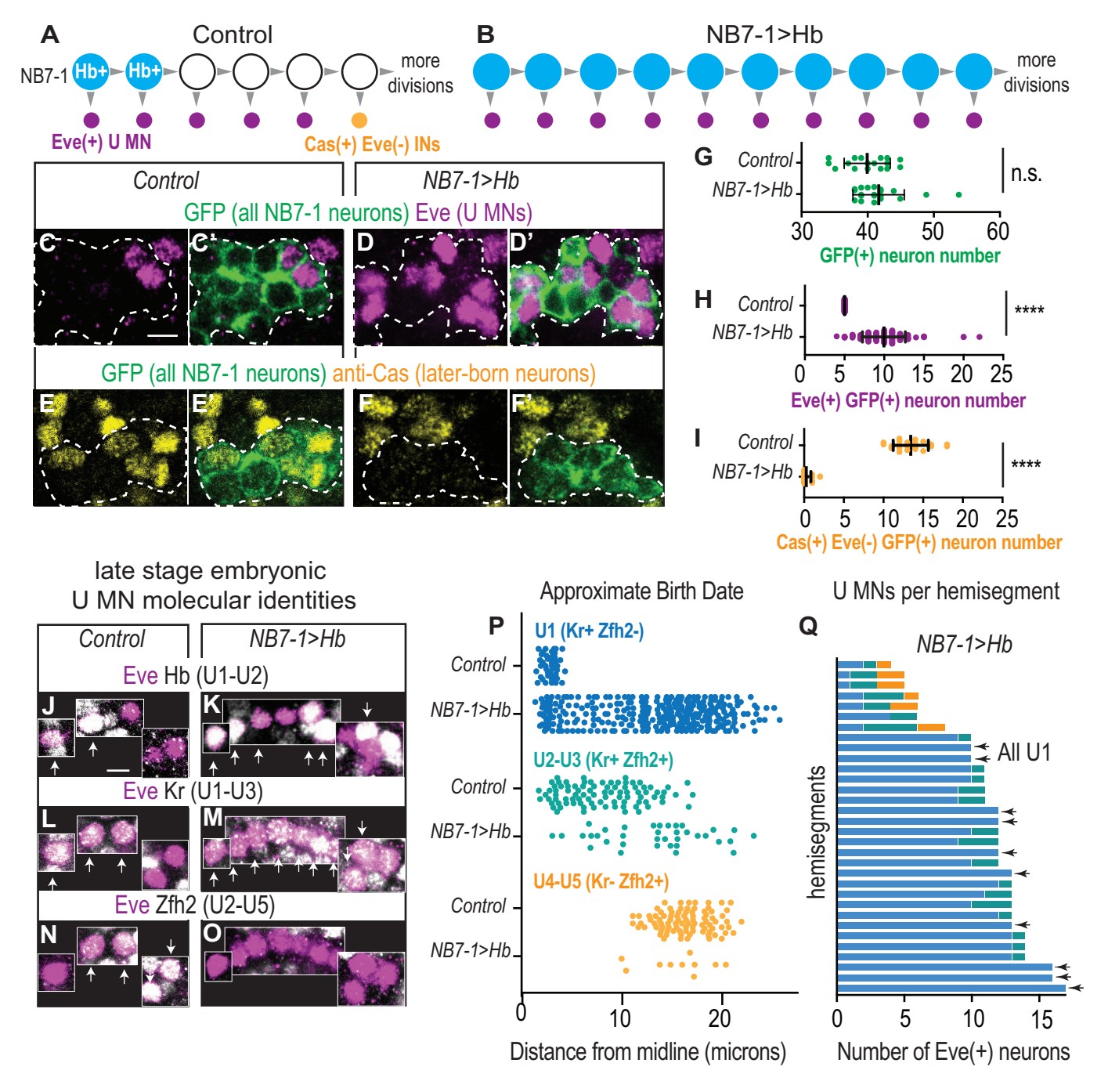

**Figure 2.** In embryos with prolonged Hb expression, U motor neurons are produced at abnormally late times in development. (A–B) Illustrations of NB7-1 lineage progression. Each gray arrowhead represents cell division. Large circles are neuroblasts, and smaller circles are neurons. In NB7−1>Hb, ectopic Eve(+) neurons (magenta) are produced when later-born Cas(+) NB7-1 interneurons (yellow) are normally produced. (C–F′) Images of late stage embryonic CNSs show distribution of Eve(+) and Cas(+) neurons. Data quantified in (G–I). GFP (+) neurons from NB7-1 are outlined in white dotted lines. (G–I) Quantification of NB7-1 neurons in Control and NB7−1>Hb. (G) Total neurons (H) Eve(+) neurons (I) Cas(+) neurons. For Controls n = 17, 23, 19 and for NB7−1>Hb n=20, 30, 14 for (G–I), respectively. (J–O) Images of embryonic molecular identity marker expression in Eve(+) cells in late stage embryonic CNSs. In NB7−1>Hb, extra Eve(+) cells are produced, most of which have U1 molecular identities. Boxes are neurons from different z-planes. Arrows indicate co-expression. (P) Quantification of distance from midline is a proxy for neuronal birth time. In NB7−1>Hb, Eve(+) cells co-labeled with U1 molecular markers are born at abnormally late times in lineage progression. Control n = 44, 88, 88, NB7−1>Hb n=301, 40, 10. (Q) Quantification of Eve(+) neuron molecular identity in single hemisegments. In NB7−1>Hb, there is hemisegment by hemisegment variability. Arrows point to hemisegments in which all Eve(+) neurons have U1 molecular identity. Color code follows (P). All images are shown in ventral view, anterior up

*Figure 2 continued on next page*

*Figure 2 continued*

and midline to the left. Scale bars represent five microns. For quantifications each dot represents single neuron cell bodies in abdominal hemisegments. Average and standard deviation are overlaid. Un-paired t-Test.' n.s.' not significant, '****'p<0.0001. In C-I, Control is *NB7-1-GAL4/+; UAS-myr-GFP/+* and NB7−1>Hb is *NB7-1-GAL4/UAS-Hb; UAS-myr-GFP/UAS-Hb*. In J-Q, Control is *w1118* and NB7−1>Hb is *NB7-1-GAL4/UAS-Hb; UAS-Hb/+*.

DOI: https://doi.org/10.7554/eLife.46089.004

The following source data and figure supplement are available for figure 2:

**Source data 1.** Source Data for *Figure 2*.

DOI: https://doi.org/10.7554/eLife.46089.006

**Figure supplement 1.** Characterization of *NB7-1-GAL4.*

DOI: https://doi.org/10.7554/eLife.46089.005

the contribution of Hb versus other birth time related factors in controlling terminal features of U motor neurons.

## When Hb expression is prolonged in NB7-1, all ectopic Eve(+) cells are motor neurons

After establishing prolonged Hb expression in NB7-1 is sufficient to generate ectopic Eve(+) neurons with embryonic U1 molecular identities, we next examine the terminal features of the ectopic neurons, and in this section we focus on motor neuron identity. In Drosophila, the gold-standard for determining if a neuron is a bona fide motor neuron is to show the neuron has a cell body in the CNS and extends a neurite out of the CNS (*Figure 3C*). This distinguishes motor neurons from interneurons whose axons do not leave the CNS, and from sensory neurons whose cell bodies are in the periphery and send axons into the CNS. We make single neuron clones in both Control and NB7−1>Hb genetic backgrounds and score for CNS exit. Regardless of genotype and birth time, every labeled Eve(+) neuron sends a neurite out of the CNS (*Figure 3A–D*). As a second, independent confirmation that ectopic Eve(+) neurons in NB7−1>Hb are motor neurons, we show a vast majority of Eve(+) neurons from NB7-1 co-stain with the pan- motor neuron marker (phosopho-Mad) (*Figure 3E–F*). We conclude that in NB7−1>Hb, Eve(+) cells are motor neurons.

## There are complete and incomplete transformations of dendrite morphology when Hb expression is prolonged in NB7-1

Next, we focus on characterizing dendrite morphology of U motor neurons in NB7−1>Hb. In Control, Hb(+) U motor neurons have contralateral dendrites that cross the midline, whereas Hb(-) U motor neurons have ipsilateral dendrites that do not cross the midline (*Figure 4A*). This raises the possibility that expression of Hb in NB7-1 is sufficient to generate contralateral U motor neuron dendrite morphology. We test this by labeling single neurons in a NB7−1>Hb genetic background and examining dendrite morphology of U motor neurons. We label 79 U motor neurons and find a majority (77%) have contralateral dendrites (*Figure 4A–D*). This demonstrates that Hb expression has a major influence on dendrite morphology, but raises the question of why in NB7−1>Hb there are a mix of ipsilateral and contralateral dendrite morphologies.

In NB7−1>Hb, a small number of U motor neurons do not express Hb (*Figure 2—figure supplement 1E*), and so an attractive idea is that these Hb(-) U motor neurons may be the U motor neurons with ipsilateral dendrites. We generate a second set of individually labeled neurons (n = 39), this time staining with Hb, and find most (79%), but not all Hb(+) U motor neurons have contralateral dendrites (*Figure 4E–F*). Because Hb(+) U motor neurons are generated from NB7-1 that expressed Hb, these data show that in NB7−1>Hb, Hb expression generates both full and partial transformations of dendrite morphology.

A recent report from Kohwi et al., showed that when Hb expression is prolonged in NB7-1, there are both full and partial transformations in U motor neuron identity, as assessed by marker gene expression (*Kohwi et al., 2013*). Specifically, they found only five Eve(+) cells are bona fide U1/U2 motor neurons (maintain endogenous Hb expression) and that the remaining Eve(+) cells have mixed molecular identity, which is due to a nuclear architecture remodeling event in the neuroblast that occurs after the fifth division. We reason that remodeling of nuclear architecture might also explain partial transformation of U motor neuron dendrite morphology. If true, we expect that in

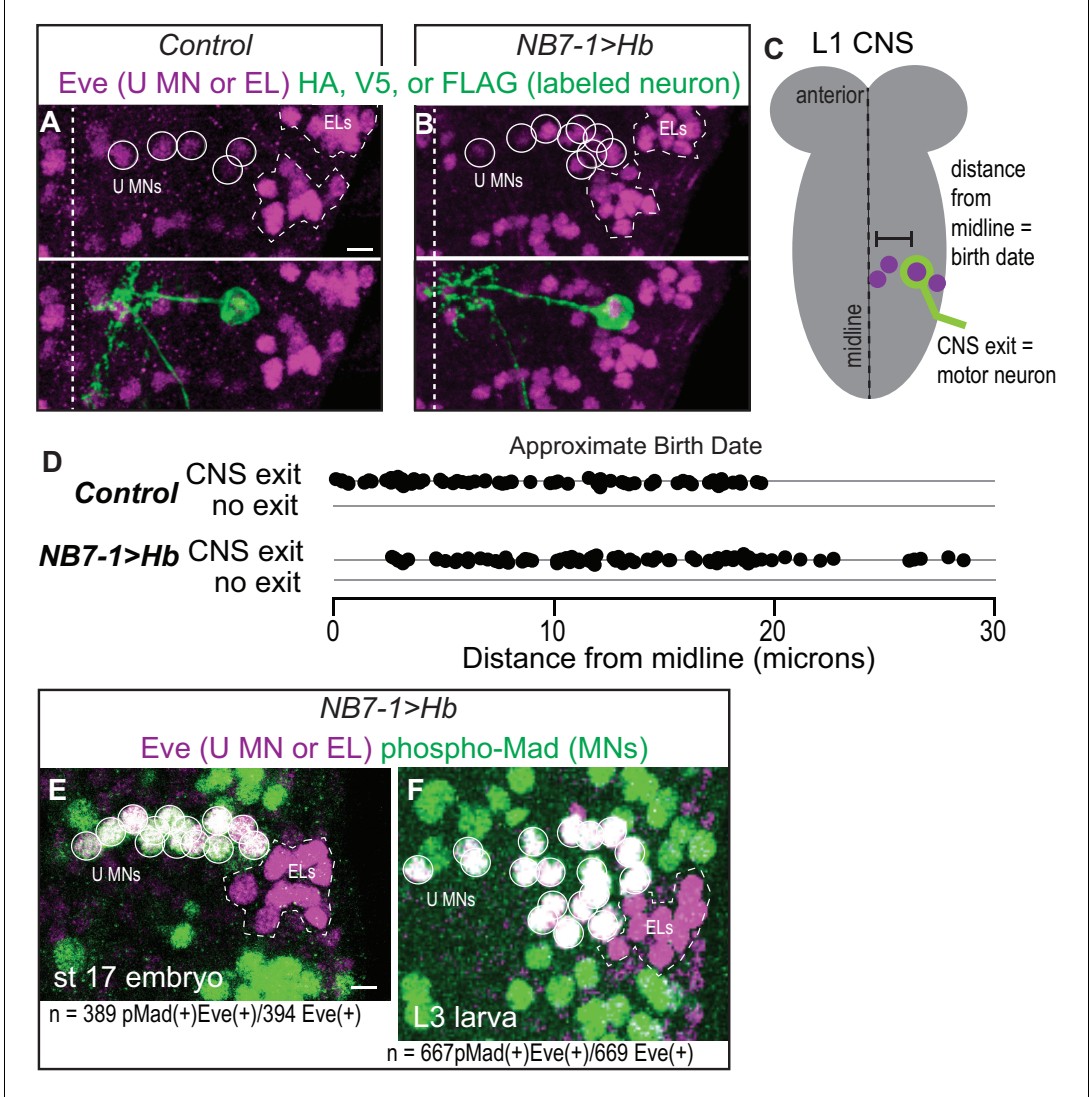

**Figure 3.** Prolonged expression of Hb in NB7-1 produces more U motor neurons. (A–B) Images of individually-labeled Eve(+) neurons in isolated first instar larvae (L1) CNSs. U motor neurons from NB7-1 are circled in white. Dotted line outlines EL interneurons (from NB3-3). Midline is dashed. (C) Illustration of isolated L1 CNS shows how data plotted in D were generated. Distance from the midline of stochastically labeled (green) motor neurons was measured. (D) Quantification of distance from midline, which is a proxy for neuronal birth time, for stochastically labeled Eve(+) cells in Control and NB7−1>Hb. Each dot represents a single neuron. Every Eve(+) neuron sends a process out of the CNS regardless of birth time. For Control n = 75, for NB7−1>Hb n=78. (E–F) Images of co-expression of Eve and the pan-motor neuron marker, pMad in NB7−1>Hb CNS of various stages. Eve(+) neurons in NB7-1 are motor neurons that survive until late larval stages. White circles outline individual U motor neurons and dotted white lines outline EL interneurons. All images are shown in ventral view, anterior up and midline to the left. Scale bars represent five microns. (A–D) Control is *hsFLP; NB7-1-GAL4/+; UAS(FRT.stop)myr::smGdP-HA, UAS(FRT.stop)myr::smGdP-V5-THSUAS(FRT.stop)myr::smGdP-FLAG/+* and NB7−1>Hb is *hsFLP; NB7-1-GAL4/ UAS Hb; UAS(FRT.stop)myr::smGdP-HA, UAS(FRT.stop)myr::smGdP-V5-THSUAS(FRT.stop)myr::smGdP-FLAG/UAS Hb*. (E–F) NB7−1>Hb is *NB7-1-GAL4/UAS-Hb; UAS-Hb/+*.

DOI: https://doi.org/10.7554/eLife.46089.007

NB7−1>Hb, Hb(+) U motor neurons born before the fifth division would have contralateral dendrites, and those born after the fifth division would have ipsilateral dendrites. We measure distance from midline of the cell body for each Hb(+) U motor neuron as a proxy for neuron birth time, plot dendrite morphology versus distance from midline, but find no association (*Figure 4G*). These data show that although prolonged expression of Hb generates partial transformations in dendrite morphology, changes in NB7-1 nuclear architecture are unlikely to account for this phenomenon.

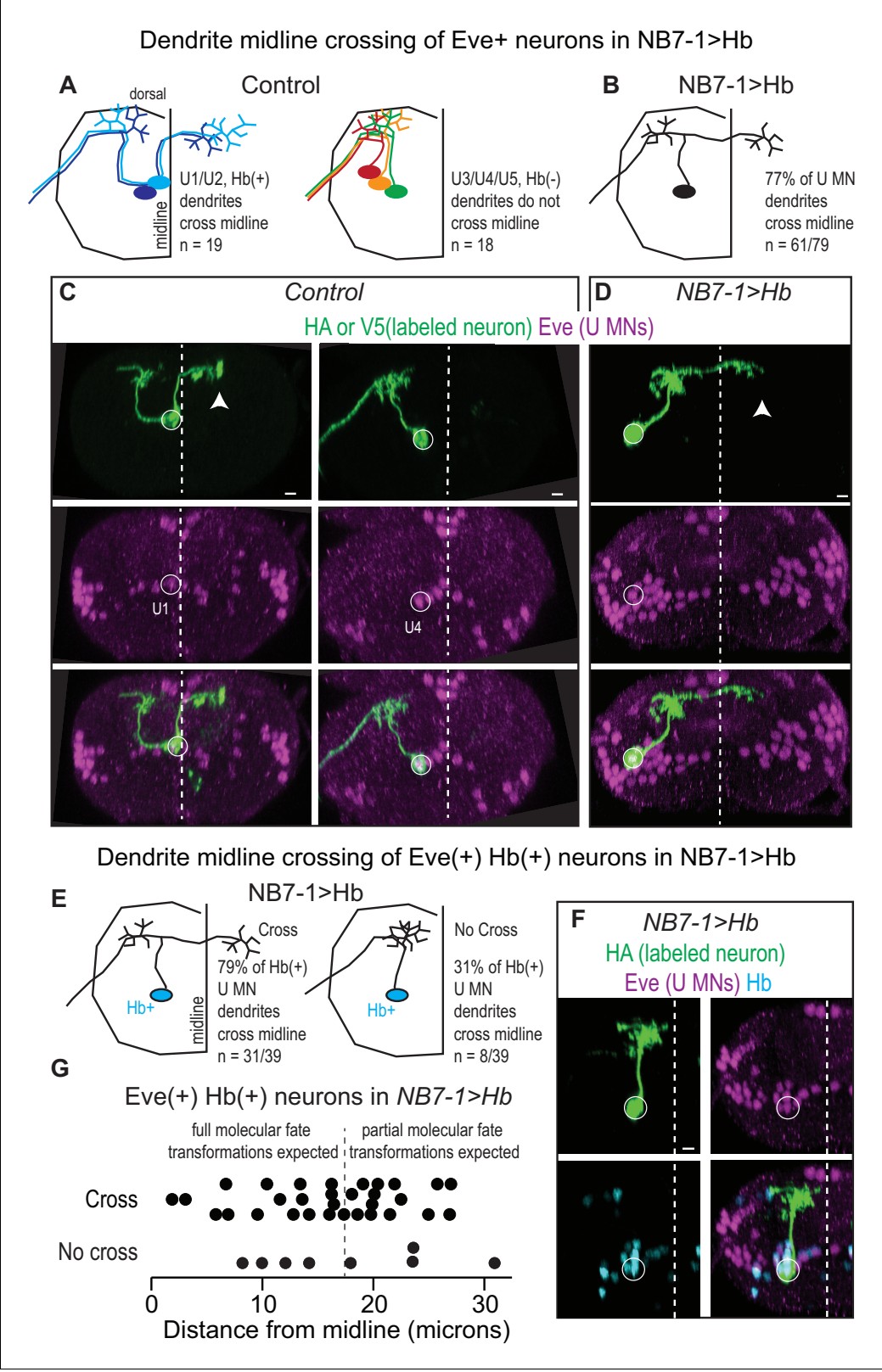

**Figure 4.** Prolonged expression of Hb in NB7-1 alters U motor neuron dendritic arbors. (**A–B**) Illustrations of U motor neuron dendrite morphology in Control and NB7−1>Hb shown in transverse view. In Control, light blue is U1, dark blue is U2, green is U3, orange is U4, and red is U5. In NB7−1>Hb, black is a Eve(+) U motor neuron. (**C–D**) Images of individually-labeled U motor neurons in the L1 CNS of Control and NB7−1>Hb. In Control, U1 has

*Figure 4 continued on next page*

*Figure 4 continued*

contralateral dendrites (white arrow) crossing the midline (white dotted line), whereas U4 has ipsilateral dendrites. In this NB7−1>Hb example, a contralaterally projecting U motor neuron is shown (white arrow). (E) Illustrations of different dendrite morphologies in NB7−1>Hb, for Hb(+) U motor neurons (blue). Most, but not all are contralateral. (G) Quantification of distance from midline for Eve(+) Hb(+) U MN neurons in L1 CNS of NB7−1>Hb. Black dotted line marks the distance from the midline for the lateral-most Eve(+) U motor neuron in Control (*W1118*), which is produced by the fifth division of NB7-1. After the fifth division of NB7-1 a nuclear remodeling event occurs. When Hb expression is prolonged in NB7-1 this remodeling event restricts the molecular fate transformations of U motor neurons born after the fifth division (*Kohwi et al., 2013*). (F) Image of Hb(+) U MN neuron dendrite morphology in L1 CNS of NB7−1>Hb. Here, in NB7−1>Hb, a Hb(+) U MN has an ipsilateral dendrite that does not cross the midline (white dotted line). All images are shown in transverse view, dorsal up, arrowheads pointing to dendrites crossing midline (white dotted line). The cell body location for labeled neurons are circled. Scale bar represents five microns. Control is *hsFLP; NB7-1-GAL4/+; UAS(FRT.stop)myr::smGdPHA, UAS (FRT.stop)myr::smGdP-V5-THS-UAS(FRT.stop)myr::smGdP-FLAG/+* and NB7−1>Hb MCFO is *hsFLP; NB7-1-GAL4/ UAS-Hb; UAS(FRT.stop)myr::smGdP-HA, UAS(FRT.stop)myr::smGdP-V5-THS-UAS(FRT.stop)myr::smGdP-FLAG/ UAS-H*b.

DOI: https://doi.org/10.7554/eLife.46089.008

---

Furthermore, these data demonstrate that embryonic U motor neuron marker gene expression does not accurately predict U motor neuron dendrite morphology.

## Prolonged expression of Hb shifts the distribution of neuromuscular synapses on dorsal muscles

Next, we ask to what extent does U motor neuron marker gene expression predict U motor neuron-to-muscle synaptic partnerships. Based on characterization of U motor neuron neuromuscular synapses in wild type larvae (*Figure 5A*) and on the average number of U motor neurons expressing markers of U1 through U5 embryonic identities in NB7−1>Hb embryos (*Figure 2*), we make the following predictions about neuromuscular synapses in NB7−1>Hb larvae (*Figure 5B*): MN3-1b and MN4-1b synapses should be absent because there are almost no U motor neurons with U4 and U5 molecular identities in NB7−1>Hb embryos. There should be MN9-1b, MN10-1b and MN2-1b branches, and of these there should be many more MN9-1b synaptic branches than MN10-1b and MN2-1b branches because there are more neurons with U1 identities than U2 and U3 identities in NB7−1>Hb embryos. To test these predictions, we assay mature circuit wiring in NB7−1>Hb larvae by labeling all neuronal membranes, including nerves and neuromuscular synaptic terminals (*Figure 5C–H*). We count the number of type 1b branches on dorsal muscles because in wild type larvae, U motor neurons make type 1b synapses on dorsal muscles. Notably, this method allows us to look at the population of motor neurons, not just individually-labeled neurons, as described above.

Our hypothesis predicts that in NB7−1>Hb larvae, MN3-1b and MN4-1b synapses should be absent. In wild type larvae, U motor neurons and other neurons exit the CNS and travel to the dorsal muscles through the Intersegmental Nerve (ISN), and U5 and U4 each have unique and reproducible nerve exit sites off of ISN before making MN4-1b and MN3-1b synapses, respectively (*Figure 5A*). In NB7−1>Hb animals there is a lack of normal nerve exits from ISN onto muscles 3 and 4 (*Figure 5B, G–H*). Thus, our data are in general agreement with our prediction, but we note two differences. First, there are no normally placed MN4-1b branches, and instead branches are reduced rather than eliminated (*Figure 5K*). In Drosophila, when a ventral motor neuron is eliminated by laser ablation, other motor neurons excessively branch to innervate the target muscle of the ablated neuron (*Chang and Keshishian, 1996*). Thus, in NB7−1>Hb, we suspect that non U-motor neurons are the source of the MN4-1b branches. To test this idea, we ablate U-motor neurons using a U motor neuron specific GAL4 line to express pro-apoptotic genes. In this manipulation, the number and placement of MN4-1b branches is indistinguishable from that seen in NB7−1>Hb (*Figure 5—figure supplement 1*). We consider this phenotype to be an example of compensation, which occurs when a muscle is under innervated. The second difference between our prediction that in NB7−1>Hb, MN4-1b and MN3-1b should be absent, is that there are often synaptic branches found dorsally on muscle 3 rather than at the normal ventral location (arrow, *Figure 5B,E–H*). These abnormally

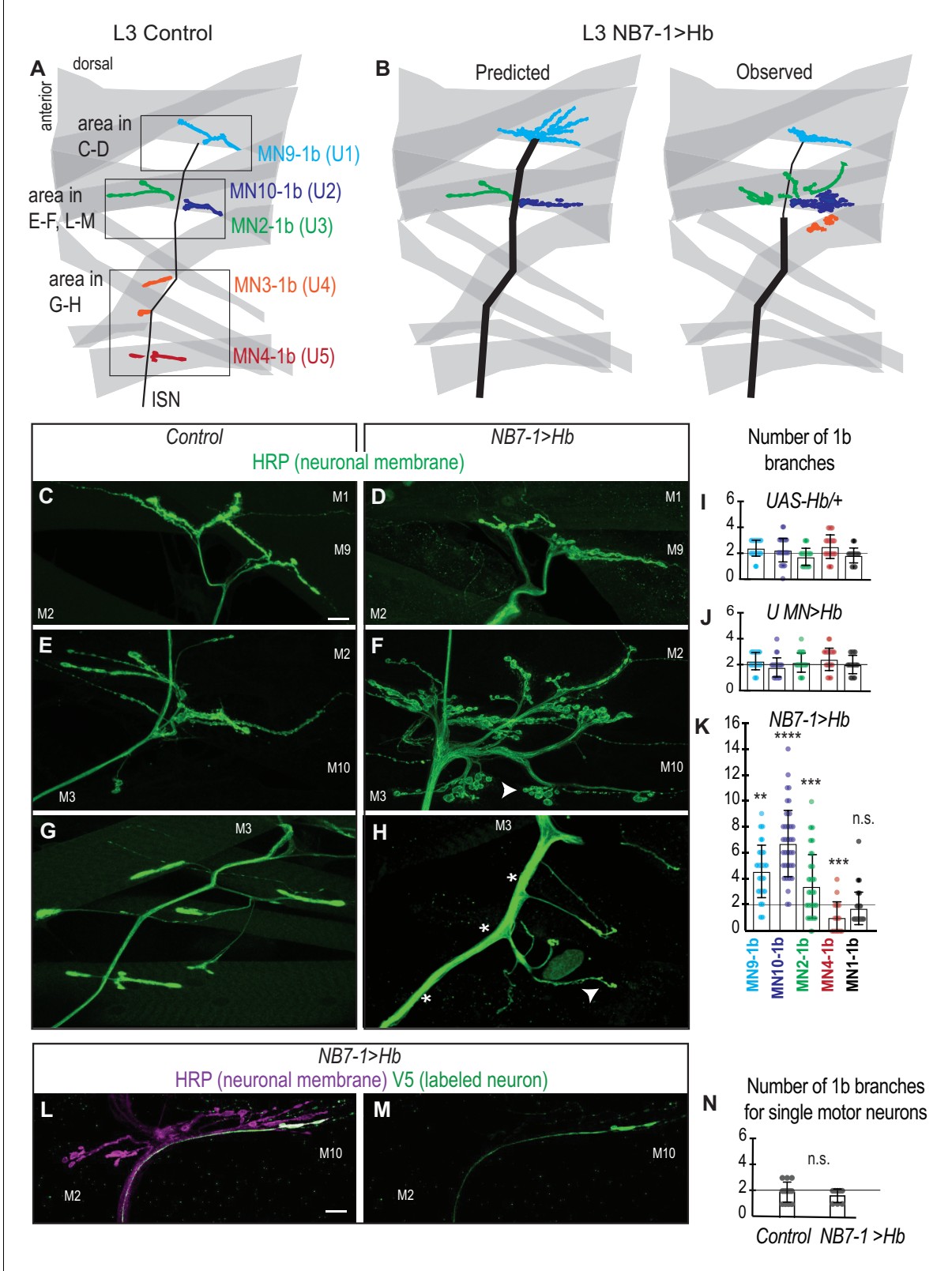

**Figure 5.** Prolonged expression of Hb in NB7-1 alters U motor neuromuscular synapses. (A–B) Illustrations of neuromuscular synapses on dorsal muscles in a L3 body wall segment. Control synapses are based on third instar larval MCFO data (see *Figure 1—figure supplement 1*). In NB7−1>Hb, predicted neuromuscular synapses are inferred from Control neuromuscular wiring patterns and from NB7−1>Hb embryonic molecular identity data (see *Figure 2*). Observed neuromuscular synapses are described in C-H. ISN is intersegmental nerve. (C–H) Images of neuronal membrane—both axons
*Figure 5 continued on next page*

*Figure 5 continued*

and neuromuscular synapses—on dorsal muscles in L3 bodywall segments. Arrow indicates branching onto M3. An asterisk * indicates missing synapses. Data quantified in (I–K). (I–K) Quantification of the number of 1b branches on L3 muscles. Color code as in (A). Line intersects the y-axis at 2. Each dot represents the number of branches onto a single muscle. (I) For UAS-Hb/+ n = 32, 35, 33, 36, 36, (J) For U MN >Hb n=11, 11, 11, 12, 11, (K) For NB7−1>Hb n=39, 46, 44, 18, 46, numbers listed from MN9-1b to MN1-1b. (L–M) Images of all neuronal membrane (magenta) on dorsal muscles in L3 bodywall segments of NB7−1>Hb with a single U motor neuron labeled (green). (N) Quantification of the number of 1b branches from individual U motor neurons on dorsal L3 muscles in Control and NB7−1>Hb. Line intersects the y-axis at two branch number. For WT n = 13, NB7−1>Hb n=8. All images are shown dorsal up, anterior left. Scale bars represent 10 microns. Control is *w1118* (C, E, G), or *hsFLP; OK6-GAL4/+; UAS-MCFO/+* (N). UAS-Hb/+ is *UAS-Hb/+, UAS-Hb/+.* U MN>Hb is *CQ2-GAL4/UAS-Hb; CQ2-GAL4/UAS-Hb.* NB7−1>Hb is *NB7-1-GAL4/UAS-Hb; UAS-Hb/+* (D–H, K) ands *hsFLP; Vglut-lexA/NB7-1-GAL4; lexAop-FRT-stop-FRT-EGFP/UAS-Hb* (L–N). For quantifications average and standard deviation are overlaid. ANOVA, corrected for multiple samples 'ns' not significant, '**' $p<0.05$, '***' $p<0.001$, '****' $p<0.0001$.

DOI: https://doi.org/10.7554/eLife.46089.009

The following source data and figure supplement are available for figure 5:

**Source data 1.** Source Data for *Figure 5*.
DOI: https://doi.org/10.7554/eLife.46089.011
**Figure supplement 1.** U Motor neuron ablation.
DOI: https://doi.org/10.7554/eLife.46089.010

located synapses are found adjacent to muscles (muscles 10 and 2 see below) that are hyper-innervated. It is likely that this phenotype occurs when a muscle has too many synapses on it, and some synapses spill over onto nearby muscles. We conclude that prolonged expression of Hb in NB7-1 can force U motor neurons to bypass the normal nerve exit point that leads to muscles 3 and 4, but that in NB7−1>Hb, there are still 1b synapses on muscles 3 and 4 likely due to compensation and spillover.

A second prediction is that in NB7−1>Hb, there should be many MN9-1b synaptic branches and fewer MN10-1b and MN2-1b synapses. We find significant increases in MN9-1b, MN10-1b and MN2-1b branch number (*Figure 5K*). Counter to our prediction, there are significantly more MN10-1b branches than MN9-1b branches (*Figure 5B–F,K*). To confirm that this unexpected phenotype is not due to an unknown role of Hb in promoting excessive branching of U motor neurons, we do two follow up experiments. First, we express Hb in post-mitotic U motor neurons and find no increase in synapse branching (*Figure 5J*). Second, we confirm that in NB7−1>Hb animals, each U motor neuron generates one or two synaptic branches, as in wild type (*Figure 5L–N*). In summary, our data reveal an unexpectedly large number of MN10-1b branches on muscle 10 (U2 target) and an unexpectedly low number of MN9-1b branches on muscle 9 (U1 target). These data show that Hb has profound effects on neuromuscular synapse formation and suggest that U motor neuron embryonic molecular identity does not accurately predict neuromuscular partner.

Our reason for examining the ability of Hb to control terminal features is to connect early developmental programs with circuit assembly. Therefore, it is important to determine the extent to which the synaptic branches in NB7−1>Hb contained functional synapses. To characterize the NB7−1>Hb neuromuscular synapses, first in fixed tissue L3 fillets, we stain for a panel of synaptic marker proteins. There is normal abundance and localization of pre-synaptic markers—for active zones (Brp), stable microtubule loops (Fustch), and synaptic vesicles (Syn)—as well as post-synaptic markers—for post-synaptic density (DLG) and neurotransmitter receptors (GluRIIA) (*Figure 6A–G*). Thus, on the cell biological level, neuromuscular synapses in NB7−1>Hb are not different from Control. In addition, we visualize neuromuscular synapse activity with a post-synaptically localized calcium sensor (*Newman et al., 2017*). In L3 larval fillets, we image spontaneous release of individual synaptic vesicles from 1b branches on muscles 1, 2, 9, and 10. In both Control and NB7−1>Hb animals, within a five-minute imaging period, we find at least one post-synaptic response in every 1b synaptic branch on every muscle imaged (Control [n = 6] and NB7−1>Hb [n = 5] *Figure 6H–I* and *Figure 6— figure supplement 1*). We conclude in NB7−1>Hb, extra branches on dorsal muscles contain functional synapses. Taken together this demonstrates that prolonged expression of the temporal transcription factor, Hb, in single neuroblast lineage has profound effects on functional circuit wiring.

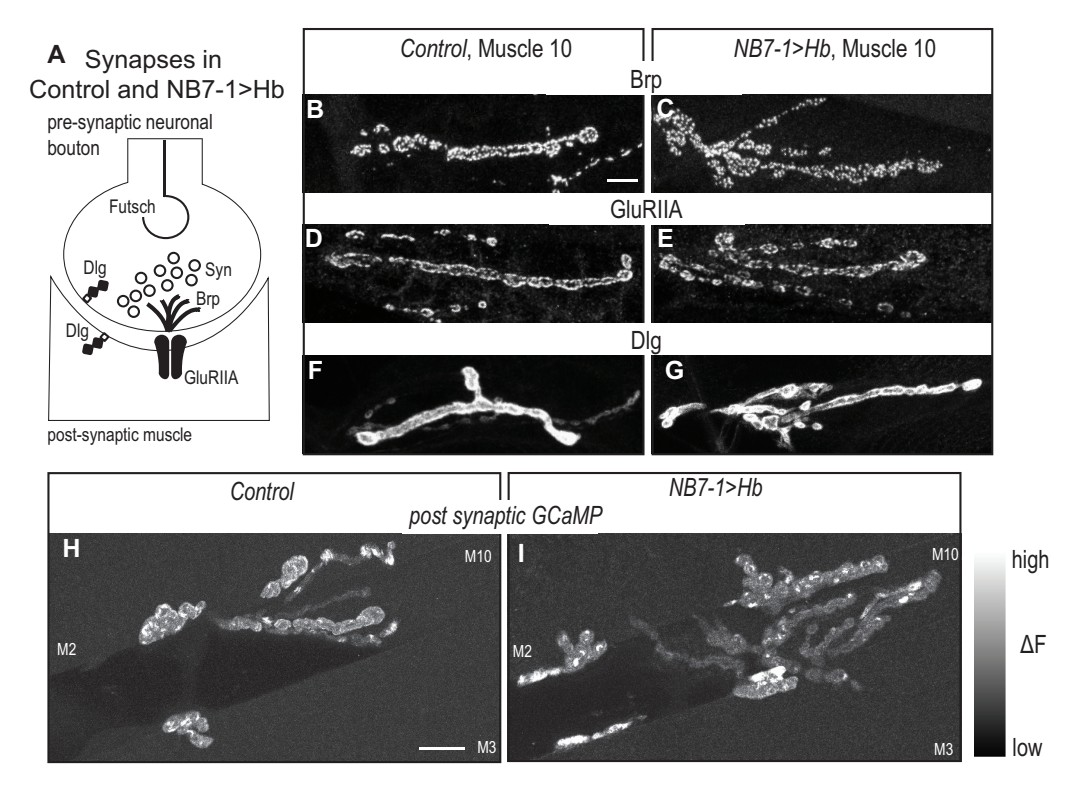

**Figure 6.** Altered synaptic arbors on dorsal muscles contain functional synapses. (A) Illustration of subcellular localization of neuromuscular synapse markers. Futsch labels microtubules; Synapsin (Syn) labels neurotransmitter filled synaptic vesicles; Brunchpilot (Brp) labels active zones; Glutamate Receptor IIA (GluRIIA) labels neurotransmitter receptors; Discs large (Dlg) is a scaffolding protein. (B–G) Images of neuromuscular synapses on L3 muscle 10. There is no difference in distribution or abundance of synaptic markers between Control (*w1118*) and NB7−1>Hb (*NB7-1-GAL4/UASHb, UAS-Hb/+*). (H–I) Images of fluorescence intensity changes in a calcium indicator of synaptic activity. GCaMP was targeted to the post-synaptic density for example (DLG in F-G). When pre-synaptic vesicles are released from active zones (Brp in B-C) post-synaptic neurotransmitter receptors respond (GluRIIA in D-E), increasing GCaMP fluorescence intensity (see *Figure 6—figure supplement 1* for details). Images show post-synaptic responses (delta F) in L3 muscles 2, 3, and 10 (M2, M3, M10) in Control (*NB7-1-GAL4/+; MHC-CD8-GCaMP6f-Sh/+*) and NB7−1>Hb (*NB7-1 GAL4/UAS Hb; MHC-CD8-GCaMP6f-Sh/UAS Hb*) over a 5 min imaging period. All images are shown dorsal up, anterior to the left. Scale bar represents 10 microns.
DOI: https://doi.org/10.7554/eLife.46089.012

The following figure supplement is available for figure 6:

**Figure supplement 1.** Calcium imaging protocol, analysis, and examples.
DOI: https://doi.org/10.7554/eLife.46089.013

## U motor neuron embryonic molecular identity and neuronal birth time do not accurately predict axonal targeting

Our neuromuscular synapse data strongly suggests that U motor neuron embryonic molecular identity does not accurately predict neuromuscular partnerships in NB7−1>Hb. We test this idea by labeling single neurons in late stage embryos, tracking axonal trajectories, and staining for embryonic U motor neuron molecular markers. Specifically we stain for Hb and Zfh2, which together distinguish U1 (Hb[+] Zfh2[-]), U2-U3 (Hb[+] Zfh2[+]), and U4-U5 (Hb[-] Zfh2[+]). We label 18 single neurons in NB7−1>Hb and find five examples where embryonic identity and axonal trajectory match as in wild type, and 13 examples where neurons with U1 identities innervate either muscle 10 or 2 (*Figure 7A–B'*). This provides direct evidence that driving Hb with NB7-1-GAL4 results in both full and partial transformations in axonal trajectory.

Next, we ask why molecular identity does not accurately predict axonal target in NB7−1>Hb. An attractive model is that in NB7−1>Hb, first-born U1 neurons reach muscle 9, make synapses on the muscle and the muscles become 'occupied', and so that later-born U1 neurons instead of synapsing on muscle 9 make synapses on muscle 10. This model predicts a correlation between the birth time

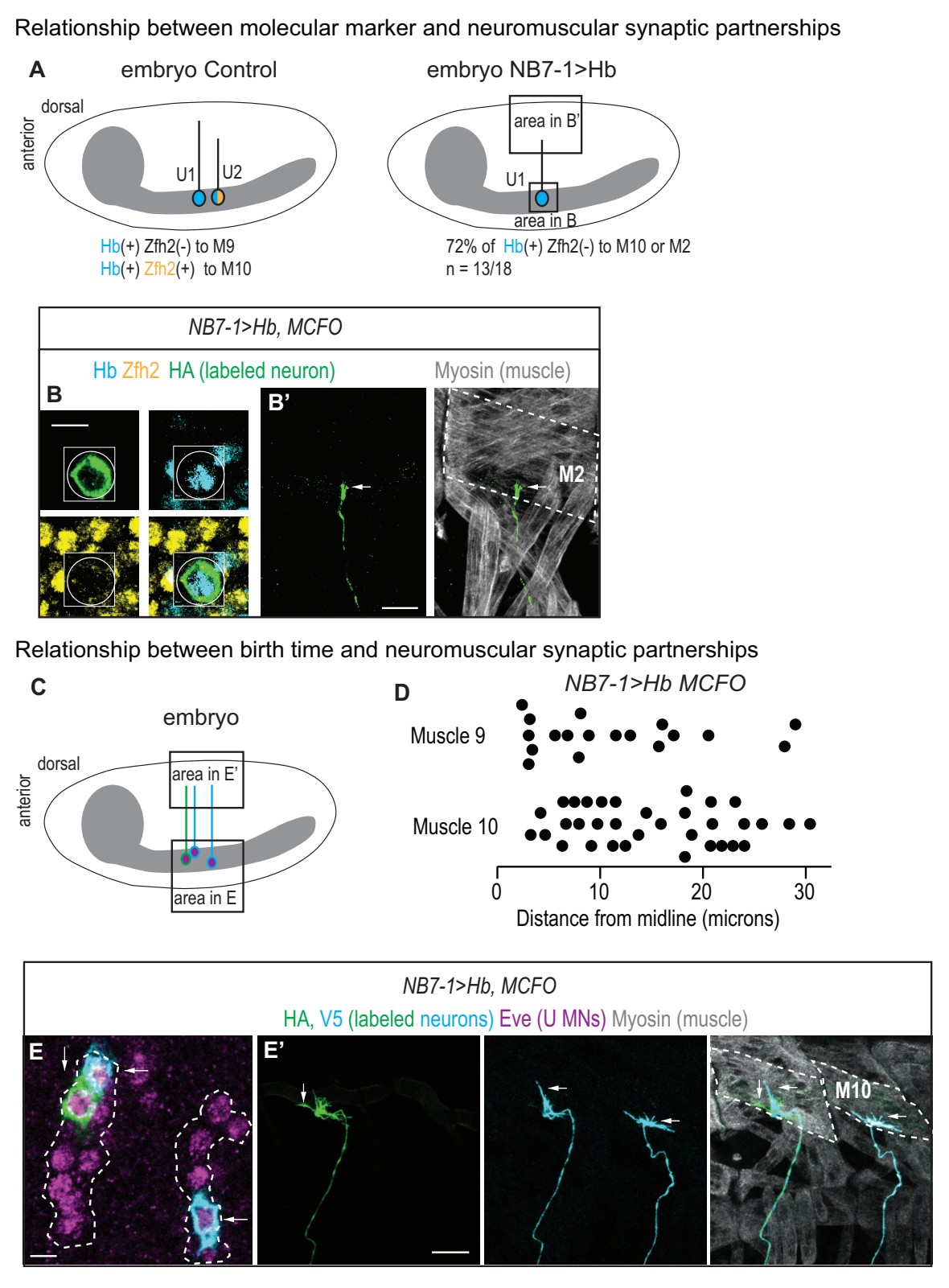

**Figure 7.** U motor neuron axonal trajectories are not accurately predicted by embryonic molecular identity or birth timing. (**A**) Illustrations of Drosophila late stage embryos, CNS is gray. In Control, Hb(+) and Zfh2(-) neurons (U1) send axons to muscle 9, and Hb(+) Zfh2(+) neurons (U2) send axons to muscle 10. Whereas in NB7−1>Hb, Hb(+) Zfh2(-) (U1) neurons often send axons to muscle 10 or muscle 2. (**B–B'**) Images of an individually-labeled U motor neuron in a late stage, whole-mount, NB7−1>Hb embryo in ventral (**B**) and lateral views (**B'**). An individual Hb(+) (cyan) Zfh2(-) (yellow) cell is

Figure 7 continued

labeled with MCFO (circled). Boxes are neurons from different z-planes. (B') The labeled U motor neuron axon projects (arrow) to muscle 2 (M2, dotted). (C) Illustration of a Drosophila late stage embryo. In NB7−1>Hb, U motor neurons at different positions along the medio-lateral axis all send axons to dorsal-most muscles, muscles 9 and 10. (D) Quantification of distance from midline, which is a proxy for neuronal birth time, for individual Eve (+) U motor neurons labeled with MCFO. Dots represent single neurons. For muscle 9 and muscle 10, n = 18, 34, respectively. (E–E') Images of a late stage, whole-mount, NB7−1>Hb embryo shown in ventral (E) and lateral (E') views. (E) There are >5 U motor neurons from NB7-1 in each hemisegment (dotted line), three of which are labeled by MCFO. (E') Axons from Eve(+) cells project to muscle 10 (M10, dotted). Vertical and horizontal arrows point to the same HA(+) and V5(+) cells, respectively (E–E'). Images in (B,E) are shown lateral up, anterior to the left. Images in (B',E'') are shown with dorsal up, anterior to the left. Scale bar represents five microns for B and E. Scale bar represents 10 microns for B' and E'. NB7−1>Hb, MCFO is *hsFLP; NB7-1-GAL4/UASHb; UAS(FRT.stop)myr::smGdP-HA, UAS(FRT.stop)myr::smGdP-V5 THSUAS(FRT.stop)myr::smGdP-FLAG/UAS-Hb.*

DOI: https://doi.org/10.7554/eLife.46089.014

of a neuron and its synaptic partner. We generate a second set of singly labeled U motor neurons (n = 52) in a NB7−1>Hb genetic background, and score both the cell body distance from midline and axonal targets. These data reveal no strong correlation between the birth time of a neuron and its axonal target (*Figure 7C–D*). They suggest that muscle occupancy as well as other time-linked factors such as changes in nuclear architecture do not strongly influence U motor neuron axon targeting in NB7−1>Hb.

## Molecular cues inherited from the neuroblast can convey wiring specificity

We were puzzled as to why prolonged expression of Hb in NB7-1 is sufficient to generate a large number of U motor neurons with U1 embryonic molecular identities, but yet these neurons do not always project axons to, and make synapses with, 9. One possibility is that in NB7−1>Hb, Hb expression may not fully transform Eve(+) neurons into bona fide U1 motor neurons. To test this, we screen all available embryonic U motor neuron molecular markers for expression in U motor neurons at late (L3) larval stages. In wild type, U1 and U2 maintain Hb expression, indicating that expression of Hb at L3 is a marker for U1/U2 temporal identity (*Figure 8A,C,D*). In NB7−1>Hb, we stain for Hb at L3, and often found two, and rarely three, bona fide Hb(+) U1/U2 motor neurons (*Figure 8E,F*), even though in late stage embryos, U motor neurons in NB7−1>Hb expressed Hb as well as other markers of U1/U2 embryonic molecular identity (*Figure 8C*). We conclude that in NB7−1>Hb, prolonged Hb in NB7-1 is only sufficient to transiently transform U motor neurons into U1 neurons.

Next, we wonder if increasing the expression level of Hb in NB7-1 increases the number of bona fide U1/U2 motor neurons at late larval stages. We increase Hb expression by raising NB7−1>Hb embryos at higher temperature or increasing transgene copy number, and confirm the efficacy of these manipulations by showing slightly higher average number of Eve(+) neurons at L3 (*Figure 8G*). However, there is absolutely no change in the number of Hb(+) Eve(+) neurons at L3 (*Figure 8G*). These data suggest the existence of a lineage intrinsic mechanism present after the second division of NB7-1 that restricts the ability of Hb to permanently transform U motor neurons into bona fide U1/U2 neurons. Furthermore, these data demonstrate that in NB7−1>Hb, neurons with embryonic U1 molecular marker gene expression are a heterogeneous population.

Next, we test the idea that heterogeneity in embryonic U motor neurons in NB7−1>Hb, can explain why some neurons with U1 molecular markers make synapses onto 9, while others make synapses elsewhere. First, to test this idea at L3, we individually-label U motor neurons, stain with Hb, and characterize neuromuscular synaptic partnerships (*Figure 8I–K*). A majority (n = 7) of the individually-labeled U motor neurons in this dataset are Hb(-) and are therefore not bona fide U1/U2 motor neurons. All of these neurons project to 10 (U2 target). There is one Hb(+) bona fide U1/U2 motor neuron, which targets the 9 (U1 target). Next, for each hemisegment, we score the number of Hb(+) U motor neurons at L3 and the number of synaptic branches on s 9 and 10. We pool data for all segments with three Hb(+) U motor neurons and pool data for all segments with two Hb(+) U motor neurons, and find a significant increase in 9 branches in hemisegments with three versus two Hb(+) neurons (*Figure 8H*). These data show that molecular differences in U motor neurons at L3 correlate with neuromuscular targeting.

Because we find a positive correlation between the number of Hb(+) U motor neurons at L3 and the number of branches on muscle 9, we next alter the number of Hb(+) U motor neurons using

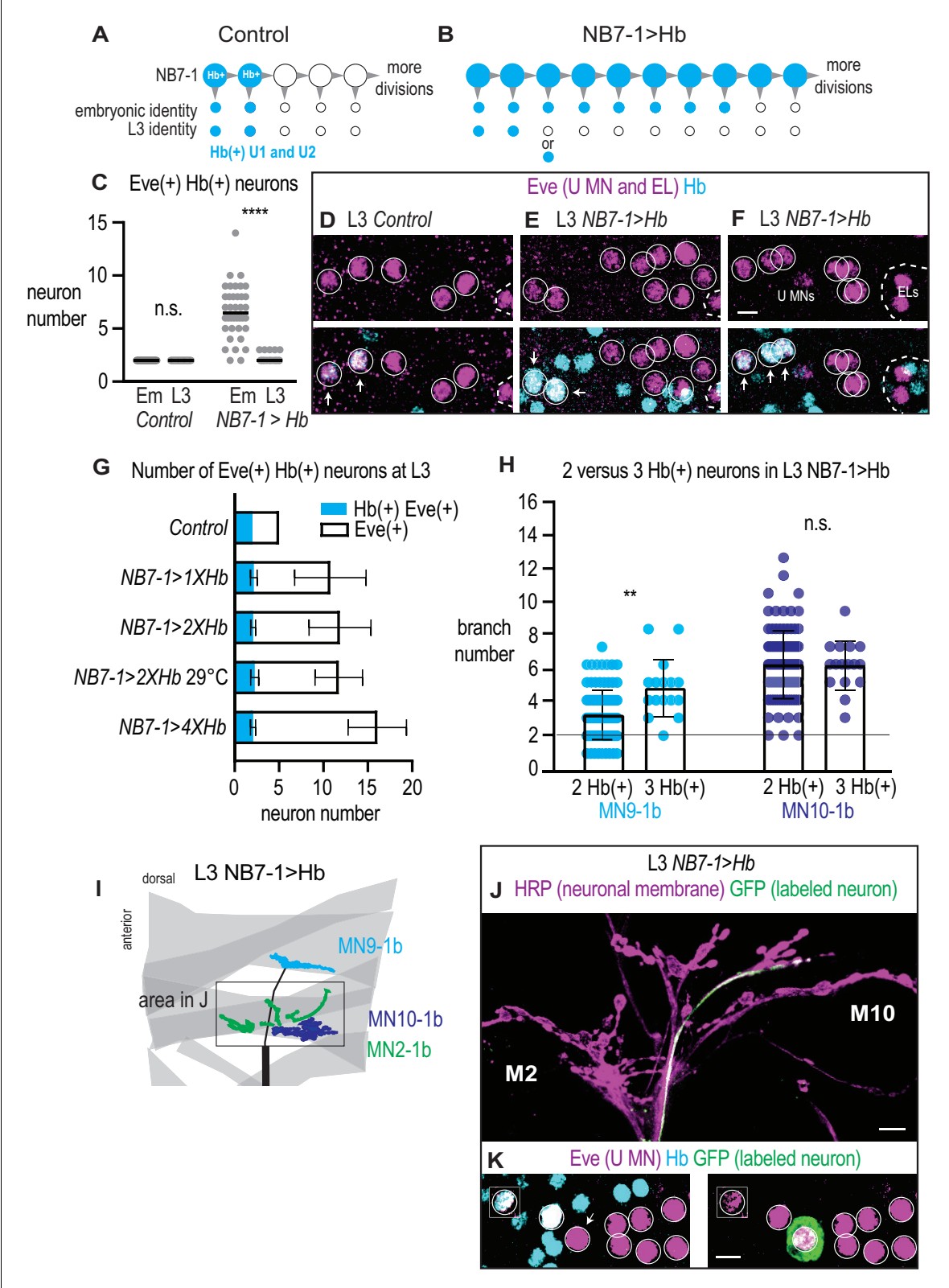

**Figure 8.** In NB7−1>Hb, U1 motor neuron identity is only transiently specified. (**A–B**) Illustrations of the divisions of NB7-1 in Control and in NB7−1>Hb. Each gray arrowhead represents cell division, large circles represent NB7-1. The top row of small circles represent U motor neurons in late stage embryos and the bottom row represent U motor neurons in L3. In Control, two neurons are Hb(+) (blue) at both stages, whereas in NB7−1>Hb many neurons are Hb(+) in embryos, but by late L3 there only two or three Hb(+) neurons. (**C**) Quantification of Eve(+) Hb(+) U motor neurons at two

*Figure 8 continued on next page*

*Figure 8 continued*

different stages: late stage embryos (Em) and late stage larvae (L3), n = 36, 28, 38, 84 from left to right. (**D–F**) Images of molecular identity, Hb(+), expression in isolated third instar larvae (**L3**) CNS. Eve(+) U motor neurons from NB7-1 are circled in white and co-localization with Hb(+) is shown (arrows). Dotted line outlines EL interneurons (from NB3-3). Data quantified in (**G**). (**G**) Quantification of Eve(+) Hb(+) U motor neuron (blue) number and total Eve(+) U motor neurons (white) neuron number at L3. Altering Hb expression level in NB7-1 by changing transgene copy number and temperature has little effect on the total number of Eve(+) neurons and has no effect on the number of Hb(+) Eve(+) neurons. For Control n = 28, NB7−1>1X Hb n = 68, NB7−1>2X Hb n = 84, NB7−1>2X Hb 29 ˚C n = 27, NB7−1>4X Hb n = 12. (**H**) Quantification of the number of 1b branches onto L3 muscle 9 and muscle 10 in NB7−1>Hb manipulations with either 2 or 3 Hb(+) Eve(+) U motor neurons. Line intersects the y-axis at two branch number. For MN9-1b 2 Hb(+) neurons n = 98, MN9-1b 3 Hb(+) neurons n = 16, MN10-1b 2 Hb(+) neurons n = 98, MN10-1b 3 Hb(+) neurons n = 16. (**I**) Illustration of neuromuscular synapses on dorsal L3 muscles in NB7−1>Hb. Black box marks area in J. A single labeled Hb(+) Eve(+) neuron projects to muscle 9, whereas seven single-labeled Hb(-) Eve(+) neurons project to muscle 10. (**J–K**) Images of a single-labeled U motor neuron in NB7−1>Hb. Eve(+) Hb(-) U motor neuron (**K**) projects to muscle 10 (**J**). Boxes are insets from a different focal plane (**K**). U motor neurons are circled. Images in (**D–F,K**) are shown in ventral view, anterior up, and midline to the left. Image in (**J**) are shown dorsal up anterior to the left. Scale bar represents five microns (**D–F, K**), and 10 microns for (**J**). Genotypes: Control is *w1118*. NB7−1>1X Hb is *NB7-1-GAL4/+; UAS-Hb/+*. NB7−1>2X Hb is *NB7-1-GAL4/UAS-Hb; UAS-Hb/+*, and NB7−1>4X Hb is *NB7-1-GAL4/UAS-Hb; NB7-1-GAL4/UAS-Hb* (for C-H). For I-K NB7−1>Hb is *hsFLP; Vglut-lexA/NB7-1-GAL4; lexAop-FRT-stop-FRT-EGFP/UAS-Hb*. For quantifications dot is number of 1b branches onto a specified muscle within a bodywall hemisegment. Average and standard deviation are overlaid. ANOVA, corrected for multiple samples. Un-paired t-tests. '**' p<0.05.

DOI: https://doi.org/10.7554/eLife.46089.015

The following source data is available for figure 8:

**Source data 1.** Source Data for *Figure 8*.

DOI: https://doi.org/10.7554/eLife.46089.016

different molecular manipulations, and then assay neuromuscular synaptic partnerships. We screen through a number of manipulations and find two that alter the number of Hb(+) U motor neurons in L3 larvae. First, when we express a wild-type Hb protein tethered to the transcriptional activation domain VP16 (NB7−1>VP16::Hb), in NB7-1 we see an increase in the number of Hb(+) neurons, reaching on average nearly 5 Hb(+) U motor neurons (*Figure 9B,D,F*). Note, that unlike NB7−1>Hb, in NB7−1>VP16::Hb the total number of Eve(+) neurons does not increase, consistent with previous publications (*Tran et al., 2010*). Second, we precociously express the Hb switching factor, Seven-up in NB7-1 and other neuroblasts, using the strong EN-GAL4 driver (EN>SVP), and we find a reduction in both the total number of U motor neurons and the number of Hb(+) U motor neurons at L3, finding on average just one Hb(+) U motor neuron per hemisegment (*Figure 9C,E,F*). Together these data show that genetic manipulations in NB7-1 can permanently alter the expression of Hb in U motor neurons.

Next, in both EN>SVP and in NB7−1>VP16::Hb, we characterize the number of Hb(+) U motor neurons versus the number of muscle 9 and muscle 10 neuromuscular synapses in L3 larvae. Within each genetic background, we pool data based on the number of L3 Hb(+) U motor neurons and then plot the number of synaptic branches on muscles 9 and 10. We find a strong correlation between the number of L3 Hb(+) U motor neurons and the number of synaptic branches on muscle 9 (*Figure 9G–M*). In contrast, there is no significant change in the number of synaptic branches onto muscle 10. Taken together these data demonstrate early manipulation of gene expression in NB7-1 is sufficient to permanently alter the molecular identity of U motor neurons and to predictably re-wire the neuronal circuits to which they contribute.

## Discussion

The existing model in the field is that temporal transcription factors are master regulators of entire developmental programs that are linked to a neuron's birth time (*Allan and Thor, 2015*). For example, it is often said that high expression of Hb in NB7-1 is both necessary and sufficient to specify U1 motor neurons fates, where fates have been taken to mean both embryonic molecular identity as well as terminal neuronal features, such as axonal trajectory (*Isshiki et al., 2001*; *Pearson and Doe, 2003*)(*Figure 10A*). From the perspective of circuit assembly, this suggests that manipulating temporal transcription factor expression in a given stem cell should predictably alter the wiring of neuronal progeny. However, recent work has introduced important refinements to the existing model, showing that prolonged expression of Hb is only sufficient to generate five bona fide U1/U2 motor neurons by embryonic molecular marker expression (*Kohwi et al., 2013*). After five divisions, NB7-1

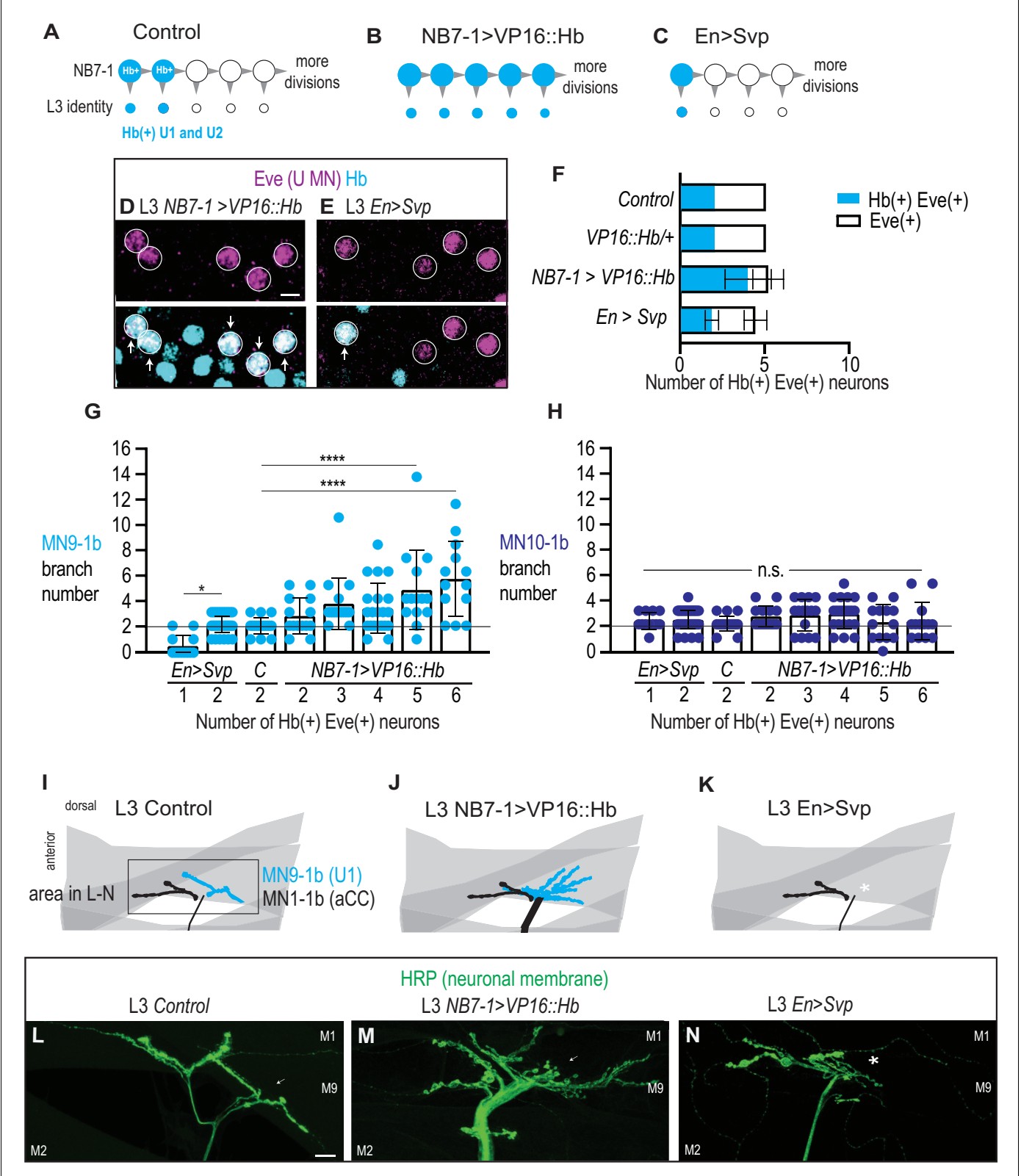

**Figure 9.** Altering the number of Hb(+) U motor neurons at L3 is correlated with synaptic partnerships. (**A–C**) Illustrations of the divisions in NB7-1 in Control, NB7−1>VP16::Hb, and En>Svp. Each gray arrowhead represents cell division, large circles represent NB7-1. Row of small circles represent U motor neuron identity at L3. In Control, two neurons are Hb(+) (blue), in NB7−1>VP16::Hb most neurons are Hb(+), and in En>Svp, only one Hb(+) remains. (**D–E**) Images of molecular identity, Hb(+), expression in isolated third instar larvae (L3) CNS, shown in ventral view, anterior up, and midline to
*Figure 9 continued on next page*

*Figure 9 continued*

the left. Eve(+) U motor neurons from NB7-1 are circled in white and co-localization with Hb(+) is shown (arrows). In NB7−1>VP16::Hb there are five Hb (+) Eve(+), whereas in En>Svp there are two (arrows). Data quantified in (F). (F) Quantification of Eve(+) Hb(+) U motor neuron (blue) number and total Eve(+) U motor neurons (white) neuron number at L3. Expressing an activated form of Hb in NB7-1 (*NB7−1>VP16::Hb*) has little effect on the total number of Eve(+) neurons but increases the number of Hb(+) Eve(+) neurons. Expressing a switching factor, Svp in NB7-1 (*En>Svp*) can decrease the total number of Eve (+) neurons by one and decrease the number of Hb(+) Eve(+) neurons by one. For Control n = 28, VP16::Hb n = 29, NB7−1>VP16::Hb n = 65, En>Svp n=70. (G–H) Quantification of the number of 1b branches onto muscle 9 and muscle 10. C. stands for Control. Line intersects the y-axis at two branch number. Number of Eve-expressing neurons are shown below each genotype. For columns left to right n = 12,44,16,14,16,21,14,13, (G) n = 12,43,17,14,15,21,13,13 (H). (I–K) Illustrations of neuromuscular synapses on L3 dorsal muscles 1 and 9. Black box marks area imaged in L-N. MN1-1b (black) comes from a non-U Motor neuron called aCC. (L–N) Images of neuronal membrane—both axons and neuromuscular synapses—on L3 dorsal muscles shown dorsal up, anterior left of Control (*W1118*) (L), NB7−1>vp16::Hb (*NB7-1-GAL4/UAS-vp16::Hb; UAS-vp16::Hb/+*) (M), and En>Svp1 (*Engrailed-GAL4/UAS-svp1*) (N). Arrow indicates branching onto muscle 9 (M9). An asterisk * indicates missing synapse. Scale bar represents five microns for E-F. Scale bar represents 10 microns for L-N. Control (*W1118*), vp16::Hb (*UAS-vp16::Hb/+; UAS-vp16::Hb/ +*), NB7−1>vp16::Hb (*NB7-1-GAL4/UAS-vp16::Hb; UAS-vp16::Hb/+*), En>Svp1 (*Engrailed-GAL4/UAS-svp1*). For quantifications, each dot represents the number of 1b branches onto a specified muscle within a bodywall hemisegment that corresponds to a hemisegment with specified number of Hb(+) neurons. Black bars are averages. Error bars represent standard deviation. ANOVA, corrected for multiple samples. Un-paired t-tests. '*' p=0.02, '****' p<0.0001, n.s. (not significant).

DOI: https://doi.org/10.7554/eLife.46089.017

The following source data is available for figure 9:

**Source data 1.** Source Data for *Figure 9*.

DOI: https://doi.org/10.7554/eLife.46089.018

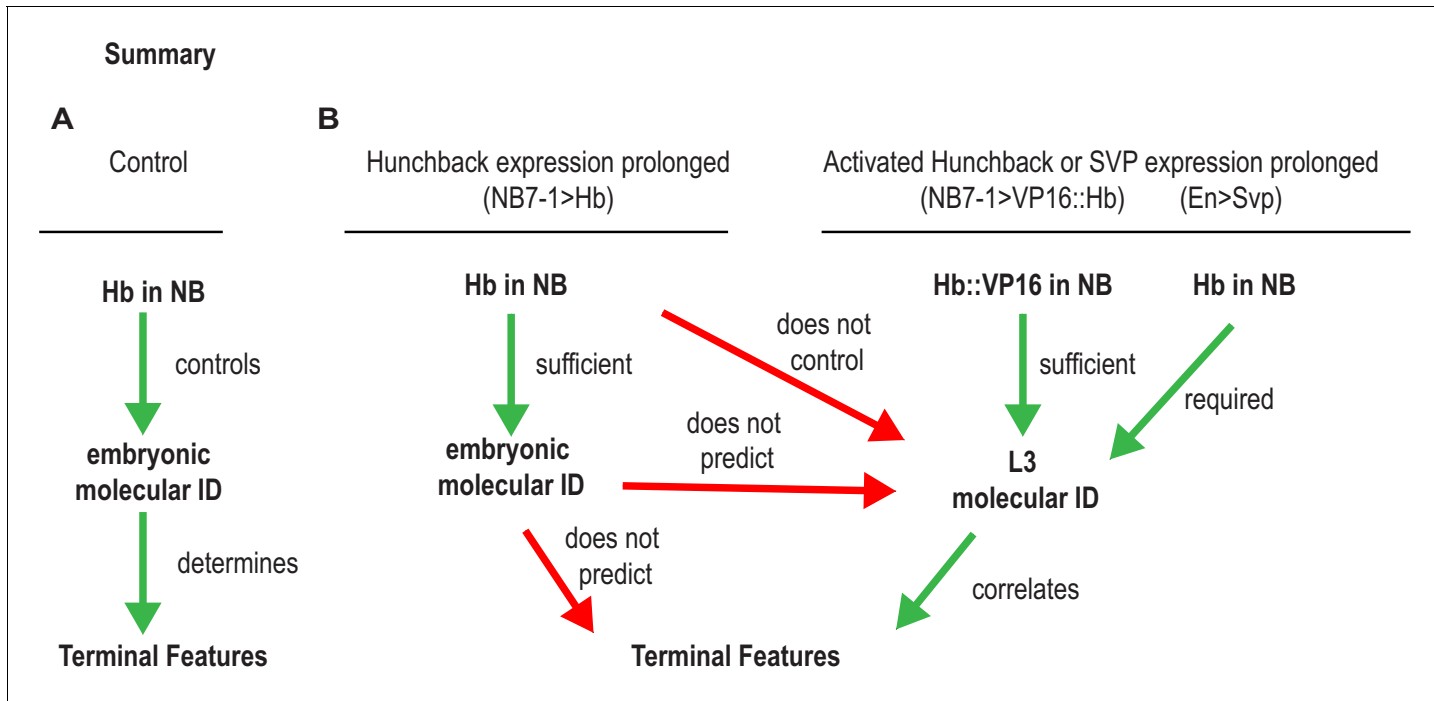

**Figure 10.** Summary of Hunchback control on circuit wiring. (A–B) Illustration summary of findings reported in this study. In Wildtype (WT) conditions, NB7-1 expresses Hunchback (Hb), Hb is involved in establishing U1/U2 motor neuron embryonic identities which was proposed to determine terminal features (dendrite morphology, axonal trajectory, and functional synaptic partnerships) (A). When Hb is overexpressed in NB7−1>Hb, Hb expression in NB7-1 is prolonged (*Figure 2*). The overwhelming majority of embryonic identities seen in motor neurons resemble U1 (*Figure 2*) however the terminal features do not entirely resemble U1 (*Figures 5–7*). We find that prolonging Hb expression in NB7-1 produces, in a majority of cases, the same number or, infrequently, one additional neuron possessing molecular identity that resembles U1 at late larval stage (L3), therefore we conclude that prolonging Hb expression in NB7-1 does not control L3 identities (*Figure 8*). Similarly, in this background, we learn that U1 embryonic identities do not predict U1 L3 identities. We find that if we overexpress an activated form of Hb (VP16::Hb) or the switching factor, Seven-up (Svp1), U1 L3 identities can be controlled. We find that L3 identities in these manipulations are correlated to terminal features of U motor neurons (*Figure 9*).

DOI: https://doi.org/10.7554/eLife.46089.019

continues to produce U motor neurons, but the molecular identity of these later-born U motor neurons is mixed. The limited production of bona fide U motor neurons has been linked to the closing of a competence window in NB7-1 mediated by a nuclear remodeling event (*Kohwi et al., 2013*). The biological relevance of this nuclear remodeling event has never been explored. Furthermore, the existence of this nuclear remodeling event highlights the idea that during neurogenesis there is a complex set of time-linked factors, including the dynamic expression of temporal transcription factors, all of which have the potential to impact neuronal wiring decisions. The objective of this study was to unravel this complexity by explicitly testing the hypothesis that the activity of the temporal transcription factor, Hb, in NB7-1 is sufficient to control terminal features of its U motor neuron progeny regardless of birth time.

## Manipulation of temporal transcription factors provides insight into neuromuscular plasticity and identifies developmental decision points for wiring specificity

Neuromuscular targeting is thought to be controlled in a step-wise fashion with motor neurons making a series of decisions that culminate in specific wiring decisions. First, a neuron must decide to exit the nerve cord, then it must decide to branch off the main nerve, and finally it must pick a specific muscle with which to synapse. For decades the Drosophila neuromuscular system has been a model to identify molecular mechanisms underlying these steps. Notably, the phenotypes we report here are among the most dramatic morphological rearrangements of neuromuscular synapses ever reported in this system (*Figures 5,9*). In general, in Drosophila larva, in mutants—for cell adhesion, axon guidance cues, and post-mitotic neuronal transcription factor genes—there is delayed rather than impaired synapse formation, which indicates there is enormous plasticity, redundancy and complexity in neuromuscular wiring decisions (*Desai et al., 1996*; *Krueger et al., 1996*; *Landgraf et al., 1999*; *Nose et al., 1994*). Here, we show that by manipulating gene expression early in neuronal development, in the neuronal stem cell, we dramatically alter neuromuscular synapse partner selection of motor neurons (*Figures 5,9*). This argues against a model in which post-mitotic motor neurons are gradually restricted to adopt specific synaptic partnerships, and instead argues that in general, when a motor neuron is born, it already 'knows' its synaptic partner. Furthermore, our data show that transcription factors acting early in neuronal development, in neuroblasts, control the ensemble of molecular effectors necessary to specify neuromuscular wiring. Thus, we identify a powerful entry point for future studies to uncover the suite of effectors controlling neuromuscular synapse selection.

The Drosophila neuromuscular system is a prominent model for studying synaptic homeostasis and plasticity. The neuromuscular wiring phenotypes documented in this study provide evidence for two types of plasticity in neuromuscular wiring, which we refer to as compensation and spillover. Compensation occurs when a muscle does not have enough innervation. In this case, neurons that do not normally innervate a muscle form extra branches to innervate a muscle that is devoid of its normal innervation (either because the motor neuron has been ablated or genetically mis-specified). We find evidence for compensation on dorsal muscles, muscle 3 and 4, similar to that which has been found for ventral muscles (*Chang and Keshishian, 1996*), showing that compensation is not a muscle- or neuron-specific phenomenon. Spillover occurs when a muscle has too much innervation. To our knowledge this type of plasticity has not yet been described because no manipulations have been identified that increase innervation on a single muscle to the degree we describe. In spillover, motor neurons that exit from the ISN at a specific branch point do not innervate the muscle target which is normally innervated by neurons exiting at that point, but instead these neurons synapse onto other nearby muscles, such that synapses form on muscle surfaces normally devoid of synapses (e.g., the top of muscle 3, *Figure 5*). Spillover phenotypes suggests that molecular cues are sufficient to send neurons out of the nerve cord, and instruct them where to branch off the main nerve, but that once neurons reach a muscle field they can innervate one of several nearby muscles. In addition, we note that the neuromuscular synaptic branches that form in NB7−1>Hb, contain functional synapses, but we know little about the function of these neurons otherwise. It will be important to undertake a detailed functional analysis of multi-innervated muscles to gain insight into homeostatic and other synaptic plasticity mechanisms.

## Hb is a limited master regulator of temporal-specific developmental programs

A widely held model in the field is that temporal transcription factors are master regulators of temporal-specific developmental programs (*Allan and Thor, 2015*; *Jacob et al., 2008*) (*Figure 10A*). This is an attractive model, but there is limited data to support it because experiments in general have assayed establishment of molecular marker gene expression in embryos, which provides limited information about an entire developmental program. We argue that understanding the role of temporal transcription factors in regulating terminal neuronal features is critically important because it provides insight into functional neural diversity and circuit wiring and because it is the end product of an entire developmental program.

One of our most surprising findings is that embryonic molecular identity markers do not accurately predict neuronal terminal features. For example, in NB7−1>Hb early stage larvae, we find some Hb(+) neurons with ipsilateral dendrites, although in wild type all Hb(+) neurons have contralateral dendrites (*Figure 4*). In late stage embryos, we find a majority of Hb(+) Zfh2(-) U1 neurons project axons to muscles 10 and 2 rather than the wild type U1 target muscle 9 (*Figure 7*). In late stage larvae, we find that Hb(-) U motor neurons often project to muscle 10, despite the fact that in wild type muscle 10 in innervated by Hb(+) U motor neurons (*Figure 8*). These data show that Hb can generate complete and partial transformations of neuronal terminal features, and demonstrate that only in certain contexts does Hb act as a master regulator of temporal-specific developmental programs.

Temporal transcription factors are known to act in the context of a dynamically changing stem cell. For example, providing pulses of Hb to NB7-1 generates extra U motor neurons, if the pluses of Hb are provided before the fifth division. If, however, Hb is provided after the fifth division, the 'competence window' closes and NB7-1 no longer generates U motor neurons (*Cleary and Doe, 2006*; *Pearson and Doe, 2003*). Furthermore, Kohwi et al., recently showed that even if Hb is provided continuously to NB7-1, and NB7-1 generates a large number of U motor neurons, only the first five express an endogenously HA-tagged Hb transgene, suggesting that only the first five U motor neurons are bona fide U1/U2 motor neurons. Our data suggest the existence of an additional competence window that closes after two divisions of NB7-1. This is because we find that although prolonged expression of Hb produces a large number of U motor neurons, most of which express markers for U1 in late stage embryos (*Figure 2*, S2), this same manipulation assayed at late larval stages produces only two bona fide Hb(+) U1/U2 motor neurons (*Figure 8*). One possibility is that if we were able to drive Hb at higher levels in NB7-1, we would generate five bona fide U1/U2 motor neurons at L3. Although our data do not completely rule out this possibility, we note that varying the expression level of Hb had no effect on the number of Hb(+) U motor neurons at L3 (*Figure 8*). We also note that expressing an artificial construct, VP16::Hb using the same NB7-1-GAL4 driver produces five Hb(+) neurons at L3 (*Figure 9*). Although, the mechanism is still unclear, this VP16::Hb manipulation demonstrates that NB7-1 is capable of generating a large number of ectopic, bona fide U1/U2 motor neurons at L3, and it controls for the NB7-1-GAL4 expression pattern. Together these data show that in response to prolonged expression of wild type Hunchback protein, NB7-1 is only able to produce two bona fide U1/U2 motor neurons.

In sum, our study adds an entirely new and unexplored level to the existing model by focusing on the role of Hb in circuit wiring rather than establishment of molecular identity, which has been intensely studied for decades (*Figure 10B*). It examines the biological relevance for a number of previously described molecular manipulations (*Kanai et al., 2005*; *Kohwi et al., 2013*; *Tran et al., 2010*). In so doing, we showed that on a cell biological level the original fate specification model was too simple. Rather than completely transform the terminal features of later-born neurons into terminal features characteristic of early-born neurons, we find that Hb can only completely control whether or not a neuron in the NB7-1 lineage is a dorsally-projecting U motor neuron or an interneuron. For all other terminal features examined, we find partial transformations. Furthermore, our data suggest that this is linked to a previously un-identified NB7-1 competence window.

One conclusion of our paper is that individual temporal transcription factors *influence*, but do not entirely determine, neuronal fate, terminal differentiation, dendritic arborization, and neuronal partners. That is to say, overexpression of one single temporal factor can only transiently change the fate of neurons, which later fail to differentiate into the correct type of neuron. A second conclusion

of our paper is that neuromuscular synaptic partner choice and circuit wiring can be established before neuronal birth by early programs of gene expression acting in neuronal stem cells. While these two conclusions may seem to contradict one another, this contradiction can be resolved by pointing out that our data, do not provide the identities of all of the genes acting in the neuronal stem cells, but point to their existence and suggest that Hb is among these genes.

## Insights into the lineage-based organization of motor circuits

In many CNS regions there are notable relationships between neuronal stem cell lineage and circuit wiring, but these relationships differ depending on the circuit configuration with a given CNS region (*Xu et al., 2014*; *Yu et al., 2009*). These observations have led to the hypothesis that lineage is an important element in circuit assembly, but that lineage-circuit relationships differ for different circuits. Little is known about lineage-circuit relationships in the motor system. In the Drosophila motor system, we recently showed that neurons from a single neuroblast, NB3-3 are organized into blocks of non-identical, yet functionally-related sets (*Wreden et al., 2017*). These early-born and late-born blocks of lineage-related interneurons are called 'temporal cohorts', defined as groups of contiguously-born neurons from a single stem cell that have similar circuit level function. To be a *bona fide* temporal cohort requires a large amount of information about individual neurons, including their exact stem cell parent, birth order, as well as a functional demonstration of circuit membership. It is important to have all of these pieces of information for a single neuron because ultimately this is critically required to identify molecular mechanisms underlying cell fate specification of neurons in a circuit.

Temporal cohorts are likely to be found in many lineages, brain regions, and organisms, and therefore could be fundamental, developmentally-based units of neuronal circuit organization. Furthermore, there are many examples of sets of neurons that partially fulfill the criteria to be considered a temporal cohort. It is likely that temporal cohorts are found in several regions of the Drosophila CNS. For example, in the Drosophila central complex, early-born neurons from Type II neuroblasts have shared dendrite and axonal projection patterns and later-born neurons have distinctive targets (*Sullivan et al., 2019*). Similarly, in mushroom bodies there is a relationship between neuronal birth time from mushroom body neuroblasts and neuronal morphology (*Zhu et al., 2003*). However, in both of these examples, the developmental origin of neurons has been mapped only to a small pool of molecularly heterogeneous neuroblasts, and so the precise stem cell origin of neurons in these CNS regions remains undefined (*Kunz et al., 2012*; *Walsh and Doe, 2017*). There are also likely to be temporal cohorts in the vertebrate spinal cord. For example, inhibitory Renshaw cells, which contribute to a motor neuron feedback circuit and Ia interneurons, which contribute to a reflex circuit are sequentially produced by progenitors in a small domain (*Benito-Gonzalez and Alvarez, 2012*; *Stam et al., 2012*). However, the precise birth time and exact stem cell parent of these spinal cord neurons is unknown. Together these studies provide evidence that developmental origin is often tightly linked to neuronal morphology and connectivity, and they illustrate the technical difficulties associated with identifying additional examples of temporal cohorts.

Here, we identify a temporal cohort in the NB7-1 lineage comprised of U motor neurons (*Figure 1*). U motor neurons are from a single neuronal stem cell, contingously-born, and because they innervate a group of synergistic muscles they share circuit membership. This finding demonstrates that temporal cohorts can be found in lineages other than NB3-3 and that temporal cohorts can be composed not just of interneurons, but also of motor neurons. Notably, the U motor neuron temporal cohort innervates the dorsal muscles, which is just one of three distinct groups of synergistic muscles in the Drosophila larval motor system (*Heckscher et al., 2012*). This raises the question of whether other groups of muscles are innervated by other temporal cohorts. Nonetheless, the identification of the U motor neuron temporal cohort supports the idea that temporal cohorts are developmental units fundamental to motor circuit organization.

One question raised by the concept of temporal cohorts is what regulates the borders of temporal cohorts, that is, how do neuronal stem cells make abrupt, rather than gradual, changes in lineage progression. For example, in NB7-1, for several divisions, motor neurons are produced, and then NB7-1 stops making motor neurons and starts making interneurons. Our data show that Hb regulates the U motor neuron temporal cohort border because when Hb expression is prolonged in NB7-1, NB7-1 produces many more functional, dorsally-projecting motor neurons at the time when NB7-1 would normally be making interneurons. Interestingly, prolonged expression of VP16::Hb does not

alter the U motor neuron temporal cohort border. Nonetheless, our data argue that rather than specify specific cell fates, a general function of temporal transcription factors is to regulate temporal cohort boundaries. They also raise the possibility that alterations in temporal transcription factor expression could be a major driver of evolution of neuronal circuits.

### Parsing the source(s) of information that control circuit wiring

In many CNS regions in many organisms there is an association between birth time and circuit membership (*Bhansali et al., 2014*; *Deguchi et al., 2011*; *Eerdunfu et al., 2017*; *Greaney et al., 2017*; *Jefferis et al., 2001*; *Kulkarni et al., 2016*; *McLean et al., 2007*; *McLean and Fetcho, 2009*; *Morrow et al., 2008*; *Osterhout et al., 2014*; *Petrovic and Hummel, 2008*; *Pujol-Martí et al., 2012*; *Tripodi et al., 2011*). The mechanisms underlying this association are likely to be varied and are still poorly understood. Identifying sources of information that control circuit wiring is critically required for understanding molecular underpinnings of circuit assembly. In complex brains, neurons are born from pools of neuronal stem cells. As stem cells divide, several factors are simultaneously changing each of which could influence the circuit membership of the resulting neuron. One factor is a dynamic environment—earlier-born neurons and later-born neurons can encounter different physical substrates or receive different signaling cues. A second type of factor is that earlier-born neurons have more time for processes like migration and axon extension. Third, over time, within the nucleus of a neuronal stem cell, the chromatin landscape can change. Finally, earlier-born and later-born neurons have different molecular profiles, which is controlled by temporal transcription factors. Determining the precise role of each of the factor associated with birth time is crucial for understanding how circuits assemble. Our data show that in NB7−1>Hb animals, even the latest-born U motor neuron can adopt terminal features characteristic of the earliest-born U motor neuron—including CNS exit, dendrite morphology, axonal targeting and functional neuromuscular synaptic partner selection. Thus, we find no limit on the ability of late-born neurons to adopt early-born features. These findings rule out many time-linked processes in circuit wiring decisions. Furthermore, we show that manipulation of temporal patterning via manipulation of gene expression in the neuroblast is sufficient to rewire locomotor circuits, indicating that circuit wiring can be established before neuronal birth by early programs of gene expression acting in neuronal stem cells.

While this manuscript was in review, a manuscript describing similar experiments was published (*Seroka and Doe, 2019*).

## Materials and methods

**Key resources table**

| Reagent type (species) or resource | Designation | Source or reference | Identifiers | Additional information |
|---|---|---|---|---|
| Genetic reagent (*D. melanogaster*) | CQ2-GAL4 | Bloomington s tock center [BL] 7468 | BDSC Cat# 7468, RRID:BDSC_7468 | |
| Genetic reagent (*D. melanogaster*) | OK6-GAL4 | Bloomington stock center [BL] 64199 | BDSC Cat# 64199, RRID:BDSC_6419 | |
| Genetic reagent (*D. melanogaster*) | hsFLP; UAS(FRT.stop) myr::smGdP-HA, UAS(FRT.stop)myr:: smGdP-V5-THS UAS(FRT.stop)myr:: smGdP-FLAG | Bloomington stock center [BL] 64085 | BDSC Cat# 64085, RRID:BDSC_64085 | |
| Genetic reagent (*D. melanogaster*) | UAS-myr-GFP | Bloomington stock center [BL] 32198 | BDSC Cat# 32198, RRID:BDSC_32198 | |
| Genetic reagent (*D. melanogaster*) | UAS-nls-GFP | Bloomington stock center [BL] 32198 | BDSC Cat# 6452, RRID:BDSC_6452 | |
| Genetic reagent (*D. melanogaster*) | UAS-Hb; UAS-HB/TM2 | Bloomington stock center [BL] 8504 | BDSC Cat# 8504, RRID:BDSC_8504 | |

*Continued on next page*

*Continued*

| Reagent type (species) or resource | Designation | Source or reference | Identifiers | Additional information |
|---|---|---|---|---|
| Genetic reagent (*D. melanogaster*) | *w1118* | Bloomington stock center [BL] 36005 | BDSC Cat# 36005, RRID:BDSC_36005 | |
| Genetic reagent (*D. melanogaster*) | *MHC-CD8-GCaMP6f-Sh* | Bloomington stock center [BL] 67739 | BDSC Cat# 67739, RRID:BDSC_67739 | |
| Genetic reagent (*D. melanogaster*) | *ac:VP16, gsb:v8v* (aka *NB7-1-GAL4*) | Minoree Kohwi (Columbia) | | |
| Genetic reagent (*D. melanogaster*) | *VGlut-lexA/cyo* | Bloomington stock center [BL] 60314 | BDSC Cat# 60314, RRID:BDSC_60314 | |
| Genetic reagent (*D. melanogaster*) | *lexA(stop.FRT) mCD8.GFP* | Bloomington stock center [BL] 57588 | BDSC Cat# 57588, RRID:BDSC_57588 | |
| Genetic reagent (*D. melanogaster*) | *UAS-VP16::Hb/-; UAS-VP16::Hb* | Chris Doe (Oregon) | | |
| Genetic reagent (*D. melanogaster*) | *UAS-svp1 1.12* | Minoree Kowhi (Columbia) | DGGR Cat# 116195, RRID:DGGR_116195 | |
| Genetic reagent (*D. melanogaster*) | *Engrailed-GAL4* | Bloomington stock center [BL] 1973 | BDSC Cat# 1973, RRID:BDSC_1973 | |
| Genetic reagent (*D. melanogaster*) | *UAS-hid,rpr* | *Zhou et al., 1997* | | |
| Antibody | rabbit anti-Eve (polyclonal) | Heckscher Lab | | 1:1000 |
| Antibody | chicken anti-GFP (polyclonal) | Aves #GFP-1020 | Aves Labs Cat# GFP-1020, RRID:AB_10000240 | 1:1000 |
| Antibody | chicken anti-V5 (polyclonal) | Bethyl #A190-118A | Bethyl Cat# A190-118A, RRID:AB_66741 | 1:300 |
| Antibody | mouse anti-HA (monclonal) | BioLegend #901501 | BioLegend Cat# 901501, RRID:AB_2565006 | 1:100 |
| Antibody | rat anti-FLAG (monoclonal) | Novus #NBP1-06712 | Novus Cat# NBP1-06712, RRID:AB_1625981 | 1:300 |
| Antibody | rat anti-Worniu (monoclonal) | Abcam #ab196362 | | 1:250 |
| Antibody | goat anti-HRP-Cy3 (polyclonal) | Jackson Immuno Research 123-165-021 | Jackson Immuno Research Labs Cat# 123-165-021, RRID:AB_2338959 | 1:300 |
| Antibody | rat anti-Runt (polyclonal) | John Rientz (UChicago) | | 1:300 |
| Antibody | guinea pig anti-Hb (polyclonal) | John Rientz (UChicago) | | 1:1000 |
| Antibody | guinea pig anti-Kruppel (polyclonal) | John Rientz (UChicago) | | 1:1000 |
| Antibody | rat anti-Zfh2 (polyclonal) | Chris Doe (Oregon) | | 1:800 |
| Antibody | rabbit anti-Castor (polyclonal) | Chris Doe (Oregon) | | 1:1000 |
| Antibody | guinea pig anti-Dbx (polyclonal) | Heather Broiher (Case Western | | 1:500 |
| Antibody | guinea pig anti-HB9 (polyclonal) | Heather Broiher (Case Western) | | 1:1000 |
| Antibody | rabbit anti-Smad3 (pMad) (polyclonal) | Abcam #52903 | Abcam Cat# ab52903, RRID:AB_882596 | 1:300 |
| Antibody | mouse anti-myosin (monoclonal) | 3E8-3D3 | DSHB Cat# 3E8-3D3, RRID:AB_2721944 | 1:100 |

*Continued on next page*

*Continued*

| Reagent type (species) or resource | Designation | Source or reference | Identifiers | Additional information |
|---|---|---|---|---|
| Antibody | mouse anti-Futsch (monoclonal) | 22C10 | DSHB Cat# 22c10, RRID:AB_528403 | 1:50 |
| Antibody | mouse anti-Brp (monoclonal) | NC82 | Creative Diagnostics Cat# DMAB9116MD, RRID:AB_2392664 | 1:50 |
| Antibody | mouse anti-DLG (monoclonal) | 4F3 | DSHB Cat# 4F3 anti-discs large, RRID:AB_528203 | 1:500 |
| Antibody | mouse anti-GluRIIA (monoclonal) | 8B4D2 | DSHB Cat# 8B4D2 (MH2B), RRID:AB_52826 | 1:25 |
| Antibody | mouse anti-En (monoclonal) | 4D9 | DSHB Cat# 4D9 anti-engrailed /invected, RRID:AB_528224 | 1:5 |
| Antibody | Cy3-HRP (polyclonal) | Jackson Immuno Research 123-025-021 | Jackson Immuno Research Labs Cat# 123-025-021, RRID:AB_2338954 | 1:400 |
| Antibody | 647-Phalliodin | Thermofisher A22287 | Thermo Fisher Scientific Cat# A22287, RRID:AB_2620155 | 1:600 |

## Fly genetics

Standard methods were used for propagating fly stocks. For all experiments, embryos and larvae were raised at 25 °C, unless otherwise noted. The following lines were used: *CQ2-GAL4* (Bloomington stock center [BL] 7468), *OK6-GAL4* (BL 64199), *hsFLP; UAS(FRT.stop)myr::smGdP-HA, UAS(FRT.stop)myr::smGdP-V5-THS UAS(FRT.stop)myr::smGdP-FLAG* (BL 64085), *UAS-myr-GFP* (BL 32198), *UAS-nls-GFP* (BL 6452), *UAS-Hb; UAS-HB/TM2* (BL 8504), *w1118* (BL 36005), *MHC-CD8-GCaMP6f-Sh* (BL 67739) *ac:VP16, gsb:v8v* (aka *NB7-1-GAL4*, gift of M. Kohwi), *VGlut-lexA/cyo* (BL 60314), *lexA (stop.FRT)mCD8.GFP* (BL 57588), *UAS-VP16::Hb/-;UAS-VP16::Hb* (gift of C. Doe), *UAS-svp1 1.12* (gift of M. Kowhi), *Engrailed-GAL4* (BL 1973) (gift of M. Kowhi), *UAS-hid,rpr* (*Zhou et al., 1997*).

## Tissue preparation

Three tissue preparations were used: Late stage whole mount embryos, in which antibody can still penetrate cuticle; isolated first instar (L1) CNSs, in which the CNS is removed from other larval tissue so that antibody reach the CNS; and third instar (L3) fillet preparations, in which the neuromuscular tissue and cuticle are dissected away from other tissue and pinned open like a book, allowing for superb immuno-labeling and visualization of larval neuromuscular synapses. For all preparations, standard methods were used for fixation in fresh 3.7% formaldehyde (Sigma-Aldrich, St. Louis, MO) [49-51]. For calcium imaging, L3 larvae expressing *MHC-CD8-GCamp6f-Sh* construct were dissected in HL3 solution containing 1.5 mM Ca2+ and 25 mM Mg2+, brains removed, body walls rinsed fresh saline, and samples imaged.

## Immunostaining

Tissue was blocked for an hour at room temperature or overnight at 4 °C in phosphate buffered saline with 2% Normal Donkey Serum (Jackson ImmunoResearch), followed by 2 hr at room temperature in primary antibodies, and 1 hr at room temperature in secondary antibodies. Primary antibodies include: rabbit anti-Eve (1:1000, Heckscher lab, see below), chicken anti-GFP (1:1000, Aves #GFP-1020), chicken anti-V5 (1:300, Bethyl #A190-118A), mouse anti-HA (1:100, BioLegend #901501), rat anti-FLAG (1:300 Novus #NBP1-06712), rat anti-Worniu (1:250 Abcam #ab196362), goat anti-HRP-Cy3 (1:300, Jackson ImmunoResearch 123-165-021) rat anti-Runt (1:300, John Rientz, UChicago), guinea pig anti-Hb (1:1000, John Rientz, UChicago), guinea pig anti-Kruppel (1:1000, John Rientz, UChicago), rat anti-Zfh2 (1:800 Chris Doe, UOregon), rabbit anti-Castor (1:1000 Chris

Doe, UOregon), guinea pig anti-Dbx (1:500 Heather Broiher, Case Western) guinea pig anti-HB9 (1:1000 Heather Broiher, Case Western), rabbit anti-Smad3 (pMad) (1:300 Abcam #52903). The following monoclonal antibodies were obtained from the Developmental Studies Hybridoma Bank, created by the NICHD of the NIH and maintained at The University of Iowa, Department of Biology, Iowa City, IA: mouse anti-myosin (1:100, 3E8-3D3), mouse anti-Futsch (1:50 22C10) mouse anti-Brp (1:50, NC82), mouse anti-DLG (1:500, 4F3), mouse anti-GluRIIA (1:25, 8B4D2), mouse anti-En (1:5, 4D9). Secondary antibodies were from Jackson ImmunoResearch and were reconstituted according to manufacturer's instructions and used at 1:400. 647-Phalliodin (1:600, Thermofisher A22287), Cy3-HRP (Jackson ImmunoResearch 123-025-021). Embryos were staged for imaging based on morphological criteria. Whole mount embryos and larval fillets were mounted in 90% Glycerol with 4% n-propyl gallate. Larvae brain preparations were mounted in DPX (Sigma-Aldrich, St. Louis, MO).

## Antibody generation

An anti-Even-skipped antibody was generated by GenScript (Piscataway, NJ). Rabbits were inoculated three times with a bacterial fusion protein containing an N-terminal 6xHis tag fused with the first 129 amino acids of Eve (Met..Arg Gln Arg). The resulting antiserum was immunopurified using the bacterial fusion protein.

## Single neuron labeling

Single U motor neurons were labeled by crossing a Multi-Color Flip Out fly line (*Nern et al., 2015*), harboring a heat-shock inducible FLP recombinase construct, to other lines of interest. For late stage embryo and L1 larval labeling, heat shock (37 ˚C for 10 min) was delivered to embryos aged 6–24 hr on apple juice caps. For L3 labeling, late stage embryos and 1$^{st}$ instar larvae were heat shocked and incubated at 25 ˚C. All CNS tissue was co-stained with Eve antibody to confirm the identity of single cell clones. In Control, U MNs were identified by their characteristic position in the CNS with U1 positioned most medial and U5 lateral.

## Image acquisition

For fixed tissue images, data were acquired on a Zeiss 800 confocal microscope with 40X oil (NA 1.3) or 63 X oil (NA 1.4) objectives, or a Nikon C2+ confocal microscope with 40X (NA 1.25) or 60X (NA 1.49) objectives. For calcium imaging, data were acquired using a Zeiss 800 confocal microscope with a 40X dipping objective (NA 1.0) using 488 nm laser power with the pinhole entirely open. Images were acquired on a Zeiss 800 confocal microscope. Images were cropped in ImageJ (NIH) and assembled in Illustrator (Adobe).

## Image analysis

*Cell body counting:* We used *NB7-1-GAL4* driving *UAS-myristoylated-GFP* to count the number of cells within the NB7-1 lineage during late stage (16-17) embryos. Embryos were co-stained with Worniu and Eve antibodies to confirm that only the NB7-1 lineage clone was labeled. To ensure the entire lineage was labeled by GFP, we scored only segments in which both an Eve(+) U1 neuron (earliest born neuron in the lineage) and NB7-1 was labeled. Since the NB7-1 lineage clone is a dense cluster of cells in x-y-z, we used a custom Fiji plug-in, that employs the *Multi point* tool to count cells.

*Type 1b branch counting:* We stained against HRP to detect the neuronal membrane and Discs large (Dlg) in L3 fillet preparations. Dlg allowed us to distinguish between type 1b and type 1s boutons. We counted contiguous stretches of Dlg staining that overlapped with HRP as a single branch. In (*Figures 8G–H*,*9*) innervation onto muscle 9 included innervation that branched off over muscle 9 but synapsed onto the most dorsal edge of muscle 10 or muscle 2.

*Calcium imaging:* X-y-z-t stacks (*Figure 6—figure supplement 1B–C*) were converted into x-y time series images using the Maximum Intensity projection function (Fiji). X-y time series images were then registered using the Register Virtual Stack Slices plug-in (Fiji) to reduce movement artifacts. Time series images were projected into two different x-y single images using either the Maximum Intensity projection function (Fiji) or Average Intensity projection function (Fiji), and then the average intensity was subtracted from the maximum to get a change in fluorescence image (*Figure 6—figure supplement 1D*).

## Statistics

Descriptive statistics: average and standard deviation are reported, except for behavior, where the average of average speed and standard error of the mean are reported. Every data point is plotted in figures. Test statistics: All data was assumed to follow a Gaussian distribution. If standard deviations were unmatched Welch's correction was applied. For numerical data in two populations, we used un-paried, two-tailed t tests. For numerical data in more than two populations, we used ordinary one-way ANOVAs with Dunnett or Games-Howell correction for multiple comparison. Analysis done using GraphPad Prism.

## Acknowledgements

Kyle Lathem, N Grace Schulz, Machiah Gill, Catarina Machado De Oliveria Catela, Xiaoxi Zhang, Marie Greaney, Edwin Ferguson, Sally Horne-Badovinac, 2018 Drosophila Neurobiology: Genes, Circuits and Behavior course, T32 GM007183 to JLM and NIH R01-NS105748, NSF GRFP (DGE-1746045) to JLM, MGCB start-up funds to ESH.

## Additional information

### Funding

| Funder | Grant reference number | Author |
|---|---|---|
| National Institute of Neurological Disorders and Stroke | R01-NS105748 | Ellie S Heckscher |
| National Institute of General Medical Sciences | T32 GM007183 | Julia L Meng |
| National Science Foundation | DGE-1746045 | Julia L Meng |
| University of Chicago | | Ellie S Heckscher |

The funders had no role in study design, data collection and interpretation, or the decision to submit the work for publication.

### Author contributions

Julia L Meng, Conceptualization, Formal analysis, Validation, Investigation, Visualization, Methodology, Writing—review and editing; Zarion D Marshall, Meike Lobb-Rabe, Investigation, Writing—review and editing; Ellie S Heckscher, Conceptualization, Resources, Supervision, Funding acquisition, Investigation, Visualization, Writing—original draft, Project administration, Writing—review and editing

### Author ORCIDs

Ellie S Heckscher (iD) https://orcid.org/0000-0001-7618-0616

### Decision letter and Author response

Decision letter https://doi.org/10.7554/eLife.46089.022
Author response https://doi.org/10.7554/eLife.46089.023

## Additional files

### Supplementary files

• Transparent reporting form
DOI: https://doi.org/10.7554/eLife.46089.020

### Data availability

All data generated or analysed during this study are included in the manuscript and supporting files.

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
