## [Decision Letter]

[Editors’ note: this article was originally rejected after discussions between the reviewers, but the authors were invited to resubmit after an appeal against the decision.]

Thank you for submitting your work entitled "How prolonged expression of Hunchback, a temporal transcription factor, re-wires locomotor circuits" for consideration by *eLife*. Your article has been reviewed by two peer reviewers, and the evaluation has been overseen by a Reviewing Editor and a Senior Editor. The following individual involved in review of your submission has agreed to reveal their identity: Tzumin Lee (Reviewer #2).

Our decision has been reached after consultation between the reviewers. Based on these discussions and the individual reviews below, we regret to inform you that your work will not be considered further for publication in *eLife*.

The reviewers appreciated the general importance of the problem and the large amount of work that went into putting this manuscript together. However, both reviewers agreed – and so do the editors – that this manuscript does not provide enough novel insight to justify further consideration at *eLife*. It was felt that this study did not add new perspectives into the already existing model. More detailed comments are found at the bottom of this message.

Reviewer #1:

This is a very carefully crafted and detailed investigation of the role of temporal identity factors in specifying neurons in the ventral nerve cord of *Drosophila*. Very extensive work from the Doe lab and its progeny has shown that neuroblasts in the ventral nerve cord undergo a series on transitions to express temporal identity factors and thus produce different types of neurons in a sequential manner.

The first gene in the series, which is expressed by all neuroblasts when they start producing neurons, is hunchback (hb) whose function has been investigated in exquisite details. hb does not specify neurons, but it is required to produce different neurons in different neuroblasts (a defining feature of temporal identity factors, as described by these authors). In the specific neuroblast NB7-1, hb is expressed in the first three temporal windows (this information is only available in the figure) but is only required to produce two motor neurons, U1 and U2 that specifically innervate two dorsal muscles.

Previous work had shown that maintenance of hb in the lineage, leads to the production of more early (hb) neurons at the expense of late (Castor) neurons. This confirmed the function of hb as an early temporal identity factor that, when over-expressed, prevents the specification of late fates: hence, more motor neurons are produced but fewer interneurons are made.

The current paper revisits these concepts in exquisite details and confirms that the assumptions about the function of hb were correct: Overall, hb over-expression leads to an increased number of U1 and U2 neurons that have much of the properties of these neurons and innervate, connect and control dorsal muscles. Later born neurons are missing. It is not clear whether or not other motor neurons (U3-U5) are transformed into earlier neurons (innervation of muscles 3 and 4 is defective, and mostly dorsal which argues for U1 and U2 replacing U4 and U5).

Although this is very careful and detailed work, it appears to be incremental as no new concept is provided. The paper does carefully prove beyond any doubt that hb over-expression prolongs the production of U1 and U2 neurons at the expense of later born neurons. This is exactly was shown before, but with fewer markers and a weaker characterization. However, this characterization was convincing enough to make a strong point. The final statement that "We conclude that prolonged expression of Hb in NB7-1 using NB7-1-GAL4 generates U motor neurons with early-born, but not late-born molecular identities" is really the exact same conclusion that is presented in earlier papers.

Reviewer #2:

This study by Heckscher's group has critically examined the ability of a classical temporal identity gene to govern progeny's terminal phenotypes as a temporal fate master gene. It extends phenotypic analysis from molecular markers to morphology and neurite targeting in the presumably complete temporal fate transformation. This long-due question should be examined for both loss- and gain-of-function conditions. As to the arbitrary focus on the gain-of-function effects, the results obtained from only one genetic manipulation show mixed phenotypes which lack single-cell resolution. One could not tell from the gross phenotypes if there co-exist fully and partially transformed neurons and if the degrees of transformation correlate with neuron birth order. Resolving the phenotypes at the single-cell level should provide further insight into the underlying mechanisms (e.g. involvement of other temporal factors or age-dependent competence).

The mixed phenotypes may include complete and partial transformation. One should conduct single-cell labeling to correlate molecular identities with morphological phenotypes. This can be achieved by multi-color flip out with a mature neuron driver (e.g. eve-GAL4 or ideally eve-LexA::VP16).

The incomplete phenotypes could possibly result from the relatively weak split-GAL4 driver. One should repeat the experiments with en-GAL4 known to induce more and exclusively U1-like neurons. One should also check eve-GAL4 to re-confirm that post-mitotic expression of ectopic Hb does not alter progeny terminal fate.

How about loss-of-Hb phenotypes at the morphology level? Given the complex morphological phenotypes observed with ectopic Hb, the authors suggest possible involvement of other early temporal factors. This possibility raises the concern over if the prospective U1/U2 neurons are completely transformed into later-born neurons in hb mutants.

A much lower penetrance in the ectopic U1/U2 molecular identity (39 out of 118 in Figure 7E way below the claimed nearly 70% being U1-like in the second paragraph of the subsection “In NB7-1>Hb embryos, a majority of U motor neurons have U1-like molecular identities”) is noted in the illustration of dendrite phenotypes.

It is hard to assess the significance of the slowed crawling behavior phenotype without first determining the roles of various dorsal muscles in larval locomotion.

---

## [Author Response]

[Editors’ note: the author responses to the first round of peer review follow.]

We would like to thank the Reviewers for their time in reviewing our manuscript and thank the Editors for allowing us to revise and resubmit to *eLife*. We have entirely re-written the text and have substantially updated the figures, briefly described here:

- Figure 1 characterizes terminal features of U motor neurons and links these to U motor neurons at single neuron level. This figure is largely the same as in our previous submission.

- Figure 2 characterizes embryonic molecular identities of Eve+ cells in the NB7-1>Hb manipulation. This data was formerly presented in Figures 5 and 6. There are minor adjustments in content, analysis, and presentation.

- Figure 3 is a new figure which demonstrates that all Eve+ cells in NB7-1>Hb are indeed motor neurons.

- Figure 4 characterizes U motor neuron dendrite morphology in NB7-1>Hb. This data was formerly presented in Figure 7. It shows that U motor neuron molecular maker gene expression does not accurately predict dendrite morphology. We update the format to make it more clear that we did two separate rounds of single neuron labeling.

- Figure 5 characterizes U motor neuron neuromuscular synapse formation in NB7-1>Hb. This data was formerly presented in Figure 3. Data on single U motor neuron labeling and expression of Hb in post-mitotic neurons has been included.

- Figure 6 demonstrates functional synapse formation of U motor neurons in NB7-1>Hb. It was formerly Figure 4, and is largely unchanged.

- Figure 7 characterizes U motor neuron axonal trajectory in NB7-1>Hb. This data was formerly presented in Figure 2. We now include a second round of single neuron labeling with molecular marker co-expression, which reveals that U motor neuron molecular identity does not accurately predict axonal trajectory in NB7-1>Hb.

- Figure 8 characterizes U motor neuron molecular identity in NB7-1>Hb at late larval stages, and looks at the relationship between the number of Hb+ Eve+ neurons in late stage larvae and neuromuscular wiring. These data are new. They reveal that Hb only transiently transforms U motor neuron identity, and this correlates with neuromuscular wiring phenotypes.

- Figure 9 manipulates U motor neuron molecular identity in NB7-1>Hb at late larval stages through two new molecular manipulations--over expression of Hb:VP16 or of SVP, and it looks at neuromuscular wiring in these backgrounds. These data are new, and show that if U motor neuron identity is permanently transformed, this is associated with expected changes in wiring on muscle 9.

- Figure 10. Summary Figure. This is a new figure.

We have re-written the Discussion to better highlight the novelty of our findings. This novelty includes:

- Adding an entirely new and unexplored level to the existing model– by examining how temporal transcription factors impact U motor neuron terminal features and circuit wiring.

- Generating important refinements to the existing model—by showing that prolonging Hb expression is not sufficient to permanently effect a neuron’s temporal identity and that embryonic molecular identity of U motor neurons cannot accurately predict U motor neuron terminal features.

- Shedding light on the lineage based organization of the motor system – by identifying U motor neurons as a temporal cohort, and by showing that prolonged expression of Hb (but not VP16:Hb) can alter temporal cohort borders.

- Identifying a neuromuscular junction phenotype that is unbelievably severe and unlike anything else reported in the literature—which provides novel entry points to understand molecular mechanisms of neuromuscular wiring and new tools to study synaptic plasticity.

*-* Providing insight into the sources of information that dictate neuromuscular wiring—by dissociating temporal patterning cues from other time linked factors.

The reviewers appreciated the general importance of the problem and the large amount of work that went into putting this manuscript together. However, both reviewers agreed – and so do the editors – that this manuscript does not provide enough novel insight to justify further consideration at eLife. It was felt that this study did not add new perspectives into the already existing model. More detailed comments are found at the bottom of this message.Reviewer #1:This is a very carefully crafted and detailed investigation of the role of temporal identity factors in specifying neurons in the ventral nerve cord of Drosophila. Very extensive work from the Doe lab and its progeny has shown that neuroblasts in the ventral nerve cord undergo a series on transitions to express temporal identity factors and thus produce different types of neurons in a sequential manner.

First, we would like to thank the reviewer for their time, and for their characterization of this manuscript as “carefully crafted and detailed”.

We agree that very extensive work from the Doe lab and others have laid out the expression of temporal identity factors, and have implicated them in producing different types of neurons. The first steps of neuronal fate specification—establishment of molecular differences in post-mitotic neurons--have been addressed by prior studies. But, a major gap in our understanding of the biology of temporal identity transcription factors has been how these factors impact terminal neuronal features and circuit wiring. This is now explicitly stated in the Introduction:

“Despite decades of work on temporal transcription factors, however, the biological relevance of most published manipulations is unknown. This is because the role of temporal transcription factors at later steps of neuronal development, establishing terminal neuronal features such as dendrite morphology, axonal trajectory, and functional synaptic partnerships are almost completely unexplored.”

We argue that terminal features are the most relevant for understanding neuronal diversity and for understanding how functional circuits are formed during neuronal development.

The first gene in the series, which is expressed by all neuroblasts when they start producing neurons, is hunchback (hb) whose function has been investigated in exquisite details. hb does not specify neurons, but it is required to produce different neurons in different neuroblasts (a defining feature of temporal identity factors, as described by these authors). In the specific neuroblast NB7-1, hb is expressed in the first three temporal windows (this information is only available in the figure) but is only required to produce two motor neurons, U1 and U2 that specifically innervate two dorsal muscles.

We clarify that Hb is only expressed before the first two divisions in NB7-1, “in NB7-1, the temporal transcription factor, Hunchback (Hb) is expressed before the first two neuroblast divisions (Figure 2A)”.

Previous work had shown that maintenance of hb in the lineage, leads to the production of more early (hb) neurons at the expense of late (Castor) neurons.

Loss of Cas from late-born NB7-1 neurons had been speculated, but not shown directly (Isshiki, 2001). We use *NB7-1-GAL4* to drive GFP in all NB7-1 progeny and co-stain with Cas to demonstrate this point directly (Figure 2G-H).

This confirmed the function of hb as an early temporal identity factor that, when over-expressed, prevents the specification of late fates: hence, more motor neurons are produced but fewer interneurons are made.

There was convincing data that confirmed the idea that Hb was an early temporal identity factor that prevented the specification of late fates. However, it did not necessarily follow that the resulting neurons were motor neurons rather than interneurons. This idea had been speculated, but not shown directly (Pearson and Doe, 2004). We provide direct evidence to support this assumption is provided in Figure 3.

The current paper revisits these concepts in exquisite details and confirms that the assumptions about the function of hb were correct: Overall, hb over-expression leads to an increased number of U1 and U2 neurons that have much of the properties of these neurons and innervate, connect and control dorsal muscles.

We agree that our data show an increase in the number of Eve+ motor neurons when Hb expression is prolonged and that most of neurons are U1-like when their molecular identity is interrogated at late embryonic stages, which is similar to what has been published in the past. But, in this submission, when we examine the same genetic background at late larval stages we find that only two Eve+ U motor neurons are Hb+ and therefore only two Eve+ motor neurons are bona fide U1/U2. This demonstrates that prolonged Hb expression transiently changes neuronal identity (Figure 8).

As the reviewer rightly points out a common assumption in the field has been that early molecular identities in post-mitotic neurons are predictive of a neuron’s mature, terminal features—such as their connectivity with specific muscles. Our data, however, provide direct evidence that assumption is not correct because U1 and U2 identity markers do not accurately predict terminal neuronal features including dendrite morphology (Figure 4), axonal trajectory (Figure 7), or neuromuscular synaptic partnerships (Figure 8).

Together these observations provide a direct challenge to the dogma in the field that Hb and other temporal transcription factors are master regulators temporal identity and of entire developmental programs.

Later born neurons are missing. It is not clear whether or not other motor neurons (U3-U5) are transformed into earlier neurons (innervation of muscles 3 and 4 is defective, and mostly dorsal which argues for U1 and U2 replacing U4 and U5).

We believe that U3-U5 are indeed missing in the NB7-1>Hb genetic background because we get similar neuromuscular wiring phenotypes when we genetically ablate all U motor neurons (Figure 5—figure supplement 1).

Although this is very careful and detailed work, it appears to be incremental as no new concept is provided. The paper does carefully prove beyond any doubt that hb over-expression prolongs the production of U1 and U2 neurons at the expense of later born neurons. This is exactly was shown before, but with fewer markers and a weaker characterization. However, this characterization was convincing enough to make a strong point. The final statement that "We conclude that prolonged expression of Hb in NB7-1 using NB7-1-GAL4 generates U motor neurons with early-born, but not late-born molecular identities" is really the exact same conclusion that is presented in earlier papers.

We respectfully disagree with the reviewer’s characterization of our work. The reviewer claims that our final statement is "We conclude that prolonged expression of Hb in NB7-1 using NB7-1-GAL4 generates U motor neurons with early-born, but not late-born molecular identities". This is not our final statement. In our first submission, this was a summary sentence in reference to Figure 6. In the very next paragraph, we stated: “However, these data argue against the assumption that a U motor neuron’s neuromuscular synaptic partner can be accurately predicted by its embryonic molecular identity”. What the reviewer suggests was our final statement, was our starting assumption. Our manuscript is not about the establishment of molecular identity. It is about how temporal patterning influences terminal neuronal features and circuit wiring. The title of the paper is “How prolonged expression of Hunchback, a temporal transcription factor, re-wires locomotor circuits”. The novelty is linking temporal patterning to circuitry not to molecular identity.

We thank the reviewer for showing us that we need to do a better job of explaining how this work is far from incremental, and introduces several new concepts. To this end, we have substantially revised the text. For example, we moved our characterization of the establishment of embryonic molecular identity from Figures 5 and 6Figures 2 and Figure 2—figure supplement 1. We explicitly discuss how our manipulation compares to previous manipulations, and describe these experiments as controls: “Because we are using a previously uncharacterized manipulation of Hb, we performed a series of control experiments to who that we elicit changes in temporal patterning similar to those described previously.” Furthermore, we entirely re-wrote the Introduction and Discussion to point out that this paper presents several new concepts. Briefly they are:

- Our data add an entirely new and unexplored level to the existing of view of temporal patterning by temporal transcription factors, and our data introduce important refinements to the existing model.

- We shed light on the lineage-based organization of the motor system.

- This work deconvolves the contribution of many simultaneously changing factors in neuronal development, all of which could contribute to circuit wiring.

- Furthermore, we identify novel genetic entry points to understand molecular mechanisms of neuromuscular wiring as well as new tools to study synaptic plasticity.

Reviewer #2:This study by Heckscher's group has critically examined the ability of a classical temporal identity gene to govern progeny's terminal phenotypes as a temporal fate master gene. It extends phenotypic analysis from molecular markers to morphology and neurite targeting in the presumably complete temporal fate transformation. This long-due question should be examined for both loss- and gain-of-function conditions.

We would like to thank Dr. Lee for taking the time to review our paper, and for pointing-out that this question is “long-due”. Our focus on gain-of-function conditions was to disrupt the association between a neuron’s birth time and its molecular identity. We have attempted to address the loss-of-function question, see below for details.

As to the arbitrary focus on the gain-of-function effects, the results obtained from only one genetic manipulation show mixed phenotypes which lack single-cell resolution.

The focus on gain-of-function effects was far from arbitrary, and we thank the reviewer for pointing out that we were unclear in our writing. We have added an entire section in the Discussion about the relationship between neuron birth timing, molecular identity and terminal neuronal features – “Parsing the source(s) of information that control circuit wiring”. For our previous submission, we conducted many genetic manipulations, but only described the results from the manipulation with the strongest effects (see below). We now include a total of eight different manipulations plus controls (Figure 2, 5, Figure 3—figure supplement 1, 8, 9). See below for details.

One could not tell from the gross phenotypes if there co-exist fully and partially transformed neurons and if the degrees of transformation correlate with neuron birth order. Resolving the phenotypes at the single-cell level should provide further insight into the underlying mechanisms (e.g. involvement of other temporal factors or age-dependent competence).

Some of the phenotypes we reported before were described a single cell resolution (see below), and we have now included this type of data for all terminal phenotypes we assay (Figures 3, 4, 7, 8). We also now examine terminal phenotypes with respect to inferred neuronal birth order (Figures 3, 4, 7). We thank the reviewer for these comments and believe adding these data has substantially improved the manuscript.

The mixed phenotypes may include complete and partial transformation. One should conduct single-cell labeling to correlate molecular identities with morphological phenotypes. This can be achieved by multi-color flip out with a mature neuron driver (e.g. eve-GAL4 or ideally eve-LexA::VP16).

- In what was formerly Figure 7, and now is Figure 4 for dendrite morphology phenotypes, we presented this type of data. We conducted single-cell labeling using multi-colored flip out, with the *NB7-1-GAL4* driver, which persists into first instar larvae. We co-labeled with the molecular identity marker, Hb, and visualized dendrite morphology. This information was presented both in Figure 7C and 7E (see below for more on 7E). and is now presented in Figure 4E-G.

- In what was Figure 2, for axon phenotypes, and now is Figure 7, we provided single neuron labeling, which lacked molecular marker co-staining. In this resubmitted version, we did a second round of single neuron labeling with Hb and Zfh2 marker expression and include this data in Figure 7A-B.

- In Figure 3, we make single neuron clones, follow their axonal trajectory and co-stain for the motor neuron marker Eve.

- For neuromuscular junction phenotypes we provide single neuron labeling with Eve and Hb co-staining in third instar larvae (Figure 8). We cannot thank you enough for this suggestion because it led us to a very interesting phenotype—hb expression only transiently generates ectopic U1-like neurons. Note, for whatever reason, single neuron clones in this genetic background are extremely infrequent, we have tried for months to make these clones. Here report on the n=9 clones we were able to make.

In all of these experiments it is clear that there exist both full and partial transformations of terminal features, with the one exception that every Eve+ cell is a motor neuron, even until late larval stages.

We also estimate neuronal birth time by measuring cell body distance from midline as an approximation of neuronal birth date. We plot single neuron terminal features versus distance from midline, but in general find no association between birth time and terminal feature (Figures 3, 4, 7).

The incomplete phenotypes could possibly result from the relatively weak split-GAL4 driver. One should repeat the experiments with en-GAL4 known to induce more and exclusively U1-like neurons.

The suggestion to use *en-GAL4* is a great idea, but because *en-GAL4* hits many lineages the embryos are very sick at later stages.

- First, we note that the levels of Hb expression we achieve are within a physiological realm. This is because loss of the Hb switching factor, Seven up extends the number of Eve+ cells to on average 7, which is similar to the number of Eve+ cells in NB7-1>Hb. Note throughout the text, we use the number of Eve+ cells as a proxy for the strength of the driver.

- Second, in many hemisegments we do induce a large number of exclusively U1-like neurons now shown in Figure 2Q.

- Third, we now manipulate the levels of Hb expression. We both increase the expression of Hb (one copy of *NB7-1-GAL4* and two copies of *UAS-Hb* raised at 29C, and two copies of both transgenes), and lower the expression of Hb (one copy of *NB7-1-GAL4* and one copy of *UAS-Hb*). Note the highest copy manipulations were extremely unhealthy. These data are reported in Figures 2—figure supplement 1, 8.

- Fourth, we express VP16:Hb which increases the number of Hb+ Eve+ U motor neurons at L3, without increasing the number of U motor neurons (Figure 9) (Tran et al., 2010).

- Finally, we raise this possibility in the Discussion, “One possibility is that if we were able to drive Hb at higher levels in NB7-1, we would generate five bona fide U1/U2 like neurons at L3.”

One should also check eve-GAL4 to re-confirm that post-mitotic expression of ectopic Hb does not alter progeny terminal fate.

To address the role of post-mitotic Hb, we used *eve-GAL4* (aka *CQ2-GAL4*) and confirmed that post-mitotic expression of Hb does not alter circuit wiring. Data can be found in Figure 5J.

How about loss-of-Hb phenotypes at the morphology level? Given the complex morphological phenotypes observed with ectopic Hb, the authors suggest possible involvement of other early temporal factors. This possibility raises the concern over if the prospective U1/U2 neurons are completely transformed into later-born neurons in hb mutants.

We tried several ways to generate Hb loss of function mutations, but because this is especially difficult because Hb is expressed at the very beginning of NB7-1 lineage progression, and NB7-1-GAL4 does not usually come on until slightly later in NB7-1 lineage progression (Figure 2—figure supplement 1).

- First, we looked at the classic, CNS only Hb mutants as in Isshiki et al., 2001. We rarely find late stage embryos in this genetic background, and those we do find have extremely abnormal CNSs, which confound our ability to assay the terminal features of U motor neurons.

- Second, we tried to use NB7-1-GAL4 to drive Hb RNAi. This achieve excellent knock down of Hb in post-mitotic neurons but not in NB7-1 itself, likely because of the timing of driver expression.

- Third, we used EN-GAL4 to express the Hb switching factor, Seven-up, which shortens the duration of Hb expression in NB7-1 (and other lineages). In this manipulation, we do not eliminate Hb expression all together, but we do get fewer Hb+ U motor neurons at late larval stages (Figure 9), which we assay for neuromuscular synapse morphology (Figure 9).

A much lower penetrance in the ectopic U1/U2 molecular identity (39 out of 118 in Figure 7E way below the claimed nearly 70% being U1-like in the second paragraph of the subsection “In NB7-1>Hb embryos, a majority of U motor neurons have U1-like molecular identities”) is noted in the illustration of dendrite phenotypes.

Thank you for this comment. The penetrance of U1/U2 is not different between experiments, but our writing did not make this clear. We re-wrote the dendrite morphology section and re-worked the figure to more clearly explain the experiments we did. We did two rounds of single-neuron labeling. One in which we only monitored morphology and one in which we monitored both morphology and molecular identity. These are now clearly labeled “Dendrite midline crossing of Eve+ neurons in NB7-1>Hb” and “Dendrite midline crossing of Eve+ Hb+ neurons in NB7-1>Hb”>

It is hard to assess the significance of the slowed crawling behavior phenotype without first determining the roles of various dorsal muscles in larval locomotion.

We removed the behavioral data.